# Robust Reward Alignment via Hypothesis Space Batch Cutting

Zhixian Xie*[1]   Haode Zhang*[2]   Yizhe Feng[2]   Wanxin Jin[1]

## Abstract

Reward design in reinforcement learning and optimal control is challenging. Preference-based alignment addresses this by enabling agents to learn rewards from ranked trajectory pairs provided by humans. However, existing methods often struggle from poor robustness to unknown false human preferences. In this work, we propose a robust and efficient reward alignment method based on a novel and geometrically interpretable perspective: hypothesis space batched cutting. Our method iteratively refines the reward hypothesis space through "cuts" based on batches of human preferences. Within each batch, human preferences, queried based on disagreement, are grouped using a voting function to determine the appropriate cut, ensuring a bounded human query complexity. To handle unknown erroneous preferences, we introduce a conservative cutting method within each batch, preventing erroneous human preferences from making overly aggressive cuts to the hypothesis space. This guarantees provable robustness against false preferences, while eliminating the need to explicitly identify them. We evaluate our method in a model predictive control setting across diverse tasks. The results demonstrate that our framework achieves comparable or superior performance to state-of-the-art methods in error-free settings while significantly outperforming existing methods when handling a high percentage of erroneous human preferences. Paper website and code can be accessed via link.

## 1. Introduction

Reinforcement learning and optimal control have shown great success in generating intricate behavior of an agent/robot, such as dexterous manipulation (Qi et al., 2023; Handa et al., 2023; Yin et al., 2023) and agile locomotion (Radosavovic et al., 2024; Tsounis et al., 2020; Hoeller et al., 2024). In these applications, reward/cost functions are essential, as they dictate agent behavior by rewarding/penalizing different aspects of motion in a balanced manner. However, designing different reward terms and tuning their relative weights is nontrivial, often requiring extensive domain knowledge and laborious trial-and-error. This is further amplified in user-specific applications, where agent behavior must be tuned to align with individual user preferences.

Reward alignment automates reward design by enabling an agent to learn its reward function directly from intuitive human feedback (Casper et al., 2023). In preference-based feedback, a human ranks a pair of the agent's motion trajectories, and then the agent infers a reward function that best aligns with human judgment. Numerous alignment methods have been proposed (Christiano et al., 2017; Ibarz et al., 2018; Hejna III & Sadigh, 2023), and most of them rely on the Bradley-Terry model (Bradley & Terry, 1952), which formalizes the rationality of human preferences. While effective, these models are vulnerable to false human preferences, where rankings may be irrational or inconsistent due to human errors, mistakes, malicious interference, or difficulty distinguishing between equally undesirable trajectories. When a significant portion of human feedback is erroneous, the learned reward function degrades substantially, leading to poor agent behavior (Lee et al., 2021a).

In this paper, we propose a robust reward alignment framework based on a novel and interpretable perspective: hypothesis space batch cutting. We maintain a hypothesis space of reward models during learning, where each batch of human preferences introduces a nonlinear cut, removing portions of the hypothesis space. Within each batch, human preferences, queried based on disagreement over the current hypothesis space, are grouped using a proposed voting function to determine the appropriate cut. This process ensures a certifiable upper bound on the total number of human preferences required. To tackle false human preferences, we introduce a conservative cutting method within each batch. This prevents false human preferences from making overly aggressive cuts to the hypothesis space and ensures provable robustness, *while eliminating the need to explicitly identify the false preferences*. Extensive experiments demonstrate that our framework achieves comparable or superior performance to state-of-the-art methods in error-free settings

---

*Equal contribution [1]Arizona State University, Tempe AZ 85283, United States. [2]Shanghai Jiao Tong University, Shanghai, China. Correspondence to: Zhixian Xie <zxxie@asu.edu>.

*Proceedings of the 42nd International Conference on Machine Learning*, Vancouver, Canada. PMLR 267, 2025. Copyright 2025 by the author(s).

while significantly outperforming existing approaches when handling high rates of erroneous human preferences.

## 2. Related Works

### 2.1. Preference-Based Reward Learning

Learning rewards from preferences for reinforcement learning agents, also named as preference-based reinforcement learning (PbRL)(Wirth et al., 2017), was early studied in (Zucker et al., 2010; Akrour et al., 2012; 2014). Early work focused on learning weights for weighted-sum feature representations. With neural network representations, scalable PbRL is developed in (Christiano et al., 2017), recently used for fine-tuning large language models (Bakker et al., 2022; Achiam et al., 2023). In those works, human preference is modelled using the Bradley-Terry formulation (Bradley & Terry, 1952). Numerous variants of PbRL have later been developed (Hejna III & Sadigh, 2023; Hejna et al., 2024; Myers et al., 2022; 2023). Recently, progress has been made to improve human data complexity. PEBBLE (Lee et al., 2021b) used unsupervised pretraining and experience re-labelling to achieve better human query efficiency. SURF (Park et al., 2022) applied data augmentation and semi-supervised learning to get a more diverse preference dataset. RUNE (Liang et al., 2022) encourages agent exploration with an intrinsic reward measuring the uncertainty of reward prediction. MRN (Liu et al., 2022) jointly optimizes the reward network and the pair of Q/policy networks in a bi-level framework. Despite the above progress, most methods are vulnerable to erroneous human preferences. As shown in (Lee et al., 2021a), PEBBLE suffers from a 20% downgrade when there exists a 10% error rate in preference data. In real-world applications of PbRL, real human users tend to make mistakes when providing feedback, due to mistakes, malicious interference, or difficulty distinguishing between equally undesirable motions.

### 2.2. Robust Learning from Noisy Labels

To tackle erroneous human preferences, recent PbRL methods draw on the methods in robust deep learning (Yao et al., 2018; Lee et al., 2019; Lukasik et al., 2020; Zhang et al., 2018; Amid et al., 2019; Ma et al., 2020; Jiang et al., 2018; Zhou et al., 2020). For example, (Xue et al., 2024) uses an encoder-decoder structure within the reward network to mitigate the impact of preference outliers. RIME (Cheng et al., 2024) and CANDERE-COACH (Li et al., 2024) handle noisy human feedback labels by filtering—RIME uses KL-divergence to filter and flip corrupted labels, while CANDERE-COACH trains a neural classifier to predict false preferences. These methods rely on prior knowledge or assumptions of true human preference distribution for false label identification. Uni-RLHF (Yuan et al.) introduces an annotation platform with a large-scale feedback dataset for reinforcement learning with human feedback,

where the quality of human feedback is achieved using accuracy thresholds and manual verification. The method may be difficult to use in practice due to the absence of ground truth and the high cost of manual inspection. (Heo et al., 2025) proposes a robust learning approach based on the mixup technique (Zhang et al., 2018), which augments preference data by generating linear combinations of preferences. While this approach eliminates the need for additional training, false human preferences could propagate through the augmentation process, ultimately degrading learning performance.

The proposed method differs from existing methods in three key aspects: (1) it requires no prior distribution assumptions for correct or false human preferences, (2) it avoids additional classification training to assess feedback quality and filter false preferences, (3) it directly updates the hypothesis space conservatively based on entire preference batches.

### 2.3. Active Learning in Hypothesis Space

Active learning based on Bayesian approaches has been extensively studied (Daniel et al., 2014; Biyik & Sadigh, 2018; Sadigh et al., 2017; Bıyık et al., 2024; Houlsby et al., 2011; Biyik et al., 2024), and many of these methods can be interpreted through the lens of hypothesis space removal. In particular, the works of (Sadigh et al., 2017; Biyik & Sadigh, 2018; Biyik et al., 2024) are closely related to ours. They iteratively update a reward hypothesis space using a Bayesian approach through (batches of) active preference queries. However, their methods are limited to reward functions that are linear combinations of preset features. More recently, (Jin et al., 2022; Xie et al., 2024) have explored learning reward or constraint functions via hypothesis space cutting, where the hypothesis space is represented as a convex polytope, and human feedback introduces linear cuts to this space. Still, these approaches are restricted to linear parameterizations of rewards/constraints.

In addition to their limitations to weight-feature parametric rewards, the aforementioned methods do not address robustness in the presence of erroneous human preference data. In this paper, we show that the absence of robust design makes these methods highly vulnerable to false preference data.

## 3. Problem Formulation

We formulate a Markov Decision Process (MDP) with a parameterized reward as $(S, A, r_{\boldsymbol{\theta}}, P_{\text{dyn}}, p_0)$, in which $S$ is the state space, $A$ is the action space, $r_{\boldsymbol{\theta}} : S \times A \times \Theta \to \mathbb{R}$ is a reward function parameterized by $\boldsymbol{\theta} \in \Theta$, $P_{\text{dyn}} : S \times A \to S$ is the dynamics and $p_0$ is the initial state distribution. The parameterization of $r_{\boldsymbol{\theta}}$ can be a neural network.

We call the entire parametric space of reward functions, $\Theta \subseteq \mathbb{R}^r$, the *reward hypothesis space*. Given any specific reward $\boldsymbol{\theta} \in \Theta$, the agent plans its action sequence $\{\boldsymbol{a}_{0:T-1}\}$

with a starting state $s_0$ maximizes the cumulated reward

$$J_{\boldsymbol{\theta}}(\boldsymbol{a}_{0:T-1}) = \mathbb{E}_{\boldsymbol{\xi}=\{s_0,a_0,\ldots,s_T\}} \sum_{t=0}^{T-1} r_{\boldsymbol{\theta}}(s_t, a_t) \quad (1)$$

over a time horizon $T$. Here, the expectation is with respect to initial state distribution, probabilistic dynamics, etc. The action and the rollout state sequence forms an agent trajectory $\boldsymbol{\xi} = \{s_0, a_0, \ldots, s_T\}$. With a slight abuse of notation, we denote below the reward of trajectory $\boldsymbol{\xi}$ also as $J_{\boldsymbol{\theta}}(\boldsymbol{\xi})$.

Suppose a human user's preference of agent behavior corresponds to an implicit reward function in the hypothesis space $\boldsymbol{\theta}_H \in \boldsymbol{\Theta}$. The agent does not know $\boldsymbol{\theta}_H$, but can query for human feedback over a trajectory pair $(\boldsymbol{\xi}^0, \boldsymbol{\xi}^1)$. Human returns a preference label $y^{\text{true}}$ for $(\boldsymbol{\xi}^0, \boldsymbol{\xi}^1)$ base on the rationality assumption (Shivaswamy & Joachims, 2015; Christiano et al., 2017; Jin et al., 2022):

$$y^{\text{true}} = \begin{cases} 1 & J_{\boldsymbol{\theta}_H}(\boldsymbol{\xi}^0) \leq J_{\boldsymbol{\theta}_H}(\boldsymbol{\xi}^1) \\ 0 & \text{otherwise} \end{cases} . \quad (2)$$

We call $(\boldsymbol{\xi}^0, \boldsymbol{\xi}^1, y^{\text{true}})$ a *true human preference*. Oftentimes, the human can give false preference due to mistakes, errors, intentional attacks, or misjudgment of two trajectories that look very similar, this leads to a *false human preference*, defined as $(\boldsymbol{\xi}^0, \boldsymbol{\xi}^1, y^{\text{false}})$, where

$$y^{\text{false}} = \begin{cases} 0 & J_{\boldsymbol{\theta}_H}(\boldsymbol{\xi}^0) \leq J_{\boldsymbol{\theta}_H}(\boldsymbol{\xi}^1) \\ 1 & \text{otherwise} \end{cases} . \quad (3)$$

**Problem statement:** We consider human preferences are queried and received incrementally, $(\boldsymbol{\xi}_k^0, \boldsymbol{\xi}_k^1, y_k)$, $k = 1, 2, 3, \ldots K$, where $y_k = y_k^{\text{true}}$ or $y_k = y_k^{\text{false}}$ and $k$ is the query index. We seek to answer two questions:

**Q1:** With all true human preferences, i.e., $y_k = y_k^{\text{true}}$ for $k = 1, 2, \ldots$, how to develop a query-efficient learning process, such that the agent can quickly learn $\boldsymbol{\theta}_H$ with a certifiable upper bound for human query count $K$?

**Q2:** When *unknown* false human preferences (3) exist, i.e., $y_k = y_k^{\text{false}}, \exists k = 1, 2, \ldots$, how to establish a provably robust learning process, such that the agent can still learn $\boldsymbol{\theta}_H$ regardless of false preferences?

## 4. Method of Hypothesis Space Batch Cutting

In this section, we present an overview of our proposed Hypothesis Space Batch Cutting (HSBC) method and address the question of **Q1**. The proofs for all lemmas and theorems can be found in Appendix D and E.

### 4.1. Overview

We start with first showing how a human preference leads to a cut to the hypothesis space. Given a true or false human preference $(\boldsymbol{\xi}^0, \boldsymbol{\xi}^1, y)$, we can always define the function

$$f(\boldsymbol{\theta}, \boldsymbol{\xi}^0, \boldsymbol{\xi}^1, y) = (1 - 2y)(J_{\boldsymbol{\theta}}(\boldsymbol{\xi}^0) - J_{\boldsymbol{\theta}}(\boldsymbol{\xi}^1)). \quad (4)$$

Here, $J_{\boldsymbol{\theta}}(\boldsymbol{\xi}^0) - J_{\boldsymbol{\theta}}(\boldsymbol{\xi}^1)$ is the reward gap between two paired trajectories. The term $1 - 2y$ is the sign determined by human label $y$. For a true human preference $(\boldsymbol{\xi}^0, \boldsymbol{\xi}^1, y^{\text{true}})$, it is easy to verify that (2) will lead to following constraints on $\boldsymbol{\theta}_H$

$$\boldsymbol{\theta}_H \in \{\boldsymbol{\theta} \in \Theta | f(\boldsymbol{\theta}, \boldsymbol{\xi}^0, \boldsymbol{\xi}^1, y^{\text{true}}) \geq 0\} \quad (5)$$

This means that a true human preference will result in a constraint (or a cut) in (5) on the hypothesis space $\Theta$, and the true human reward $\boldsymbol{\theta}_H$ satisfies such constraint.

With the above premises, we present the overview of the method of Hypothesis Space Batch Cutting (HSBC) below.

---

**HSBC Algorithm**

With initial hypothesis space $\Theta_0 \subseteq \Theta$, perform the following three steps at iteration $i = 0, 1, 2, \ldots I$

**Step 1**: [Hypothesis sampling] Sample an ensemble $\mathcal{E}_i$ of reward parameters $\boldsymbol{\theta}$s from the latest hypothesis space $\Theta_i$ for trajectory generation.

**Step 2**: [Disagreement-based human query] Use the ensemble $\mathcal{E}_i$ to generate $N$ trajectory pairs based on disagreement for active human query and obtain a batch of human preferences $\mathcal{B}_i = \{(\boldsymbol{\xi}_{i,j}^0, \boldsymbol{\xi}_{i,j}^1, y_{i,j})\}_{j=1}^N$.

**Step 3**: [Hypothesis space cutting] Calculate the constraint set (batch cuts) $\mathcal{C}_i$:

$$\mathcal{C}_i = \{\boldsymbol{\theta} | f(\boldsymbol{\theta}, \boldsymbol{\xi}_{i,j}^0, \boldsymbol{\xi}_{i,j}^1, y_{i,j}) \geq 0, j = 1, \ldots, N\}. \quad (6)$$

and update the hypothesis space by

$$\Theta_{i+1} = \Theta_i \cap \mathcal{C}_i, \quad (7)$$

---

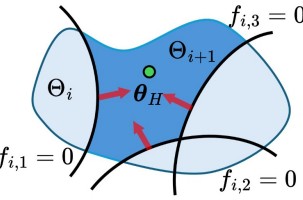

Figure 1: Illustration of update from $\Theta_i$ to $\Theta_{i+1}$. Three constraints are induced from a preference batch of size 3, with the simplified notation $f_{i,j}(\boldsymbol{\theta}) = f(\boldsymbol{\theta}, \boldsymbol{\xi}_{i,j}^0, \boldsymbol{\xi}_{i,j}^1, y_{i,j})$. Red arrows are the directions of constraints, i.e., the region of $\{\boldsymbol{\theta} | f_{i,j}(\boldsymbol{\theta}) \geq 0\}$. New $\Theta_{i+1}$ is the regions in $\Theta_i$ satisfying all constraints.

As shown in Fig. 1, the intuitive idea of the HSBC algorithm is to maintain and update a hypothesis space $\Theta_i$, starting from an initial set $\Theta_0$, based on batches of human preferences. Each batch of human preferences (**Step 2**) is actively queried. The human preference batch is turned into a batch cut (**Step 3**), removing a portion of the current hypothesis space, leading to the next hypothesis space.

From (7) in (**Step 3**), it follows $\Theta_0 \supseteq \Theta_1 \supseteq \Theta_2 \supseteq \ldots$. This means that the hypothesis space is not increasing. More importantly, we have the following assertion, saying the human reward is always contained in the current hypothesis if all human preferences are true in a sense of (2).

**Lemma 4.1.** *If all human preferences are true, i.e., $y_{i,j} = y_{i,j}^{true}$, $\forall i, j$, and $\boldsymbol{\theta}_H \in \Theta_0$, then following HSBC Algorithm, one has $\boldsymbol{\theta}_H \in \Theta_i$ for all $i = 1, 2, ..., I$.*

### 4.2. Voting Function for Human Preference Batch

In the HSBC algorithm, directly handling batch cut $\mathcal{C}_i$ and hypothesis space $\Theta_i$ can be computationally difficult, especially for high-dimensional parameter space, e.g., neural networks. Thus, we next introduce the concept of "voting function" on the hypothesis space.

Specifically, we define a voting function for a batch of human preferences $\mathcal{B}_i = \{(\boldsymbol{\xi}_{i,j}^0, \boldsymbol{\xi}_{i,j}^1, y_{i,j})\}_{j=1}^N$ as follows

$$V_i(\boldsymbol{\theta}) = \sum_{j=1}^N \mathbf{H}(f(\boldsymbol{\theta}, \boldsymbol{\xi}_{i,j}^0, \boldsymbol{\xi}_{i,j}^1, y_{i,j})), \qquad (8)$$

where $H$ is the Heaviside step function defined as $\mathbf{H}(x) = 1$ if $x \geq 0$ and $\mathbf{H}(x) = 0$ otherwise. Intuitively, any point $\boldsymbol{\theta} \in \Theta$ in the hypothesis space will get a "+1" vote if it satisfies $f(\boldsymbol{\theta}, \boldsymbol{\xi}_{i,j}^0, \boldsymbol{\xi}_{i,j}^1, y_{i,j}) \geq 0$ and "0" vote otherwise. Therefore, $V_i(\boldsymbol{\theta})$ represents the total votes of any point in the hypothesis space after the $i$th batch of human preferences.

Thus, $\boldsymbol{\theta} \in \mathcal{C}_i$ in (6) if and only if $V_i(\boldsymbol{\theta}) = N$ because $\boldsymbol{\theta}$s need to satisfy all $N$ constraints $f(\boldsymbol{\theta}, \boldsymbol{\xi}_{i,j}^0, \boldsymbol{\xi}_{i,j}^1, y_{i,j}) \geq 0$ for $j = 1, 2, ..., N$. As the number of votes in (8) only takes integers, the batch cut $\mathcal{C}_i$ can be equivalently written as:

$$\mathcal{C}_i = \{\boldsymbol{\theta} | V_i(\boldsymbol{\theta}) \geq N - 0.5\}. \qquad (9)$$

Furthermore, the indicator of batch cut $\mathcal{C}_i$ can be written as:

$$\mathbb{1}_{\mathcal{C}_i}(\boldsymbol{\theta}) = \mathbf{H}(V_i(\boldsymbol{\theta}) - N + 0.5). \qquad (10)$$

Following (7) in the HSBC, since $\Theta_i = \Theta_0 \cap \bigcap_{k=0}^{i-1} \mathcal{C}_k$, we can define the indicator function of $\Theta_i$ as:

$$\mathbb{1}_{\Theta_i}(\boldsymbol{\theta}) = \mathbb{1}_{\Theta_0}(\boldsymbol{\theta}) \prod_{k=0}^{i-1} \mathbb{1}_{\mathcal{C}_k}(\boldsymbol{\theta}), \quad i \geq 1 \qquad (11)$$

with $\mathbb{1}_{\Theta_0}(\boldsymbol{\theta})$ be the indicator function of initial hypothesis space $\Theta_0$. With the indicator representation of $\Theta_i$ in (11), we will show $\boldsymbol{\theta}$ can be sampled from $\Theta_i$ (**Step 1**) using (11).

### 4.3. Disagreement-Based Query and its Complexity

In the HSBC algorithm, not all batch cuts have similar effectiveness, i.e., some batch cut $\mathcal{C}_i$ can be redundant without removing any volume of hypothesis space, leading to an unnecessarily human query. To achieve effective cutting, each trajectory pair $(\boldsymbol{\xi}_{i,j}^0, \boldsymbol{\xi}_{i,j}^1, y_{i,j})$ should attain certain disagreement on the current hypothesis space $\Theta_i$, before sent to the human for preference query.

Specifically, to collect human preference $\mathcal{B}_i$, we only query human using trajectory pairs $(\boldsymbol{\xi}_{i,j}^0, \boldsymbol{\xi}_{i,j}^1, y_{i,j})$s that satisfy

$$\exists \, \boldsymbol{\theta}_1, \boldsymbol{\theta}_2 \in \Theta_i, f(\boldsymbol{\theta}_1, \boldsymbol{\xi}_{i,j}^0, \boldsymbol{\xi}_{i,j}^1, y_{i,j}) \geq 0$$
$$\text{and } f(\boldsymbol{\theta}_2, \boldsymbol{\xi}_{i,j}^0, \boldsymbol{\xi}_{i,j}^1, y_{i,j}) \leq 0, \; \forall y_{i,j} \quad (12)$$

Intuitively, the disagreement-based query can ensure at least some portion of the hypothesis space is removed by the constraint $f(\boldsymbol{\theta}, \boldsymbol{\xi}_{i,j}^0, \boldsymbol{\xi}_{i,j}^1, y_{i,j}) \geq 0$, no matter what preference label $y_{i,j}$ human will provide. A geometric illustration is given in Fig. 2, with simplified notation $f_{i,j}(\boldsymbol{\theta}) := f(\boldsymbol{\theta}, \boldsymbol{\xi}_{i,j}^0, \boldsymbol{\xi}_{i,j}^1, y_{i,j})$. The above disagreement-based query strategies are also related to prior work (Christiano et al., 2017; Lee et al., 2021b).

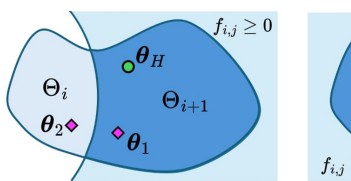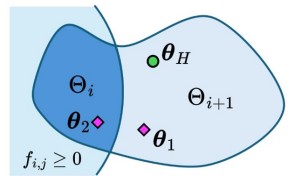

Figure 2: Illustration of disagreement-based preference query. Regardless of the human actual preference label (which only determines which side is to cut), the disagreement condition will always guarantee at least a portion of the hypothesis space is removed.

With the disagreement-based preference query in each batch, the following assertion provides an upper bound on the human query complexity of the HSBC algorithm given all true human preferences.

**Theorem 4.2.** *In the HSBC algorithm, suppose all true human preferences. Let $P(\boldsymbol{\xi}^0, \boldsymbol{\xi}^1, y^{true})$ be an unknown distribution of true human preferences. Define $\mathrm{err}(r_{\boldsymbol{\theta}}) = P(f(\boldsymbol{\theta}, \boldsymbol{\xi}^0, \boldsymbol{\xi}^1, y^{true}) < 0)$, which is the probability that $r_{\boldsymbol{\theta}}$ makes different predictions than human. For any $\epsilon, \delta$,*

$$P(\mathrm{err}(r_{\boldsymbol{\theta}}) \leq \epsilon) \geq 1 - \delta \qquad (13)$$

*holds with human query complexity:*

$$K = NI \leq O(\zeta(d \log \zeta + \log \left( \log \frac{1}{\epsilon} / \delta \right)) \log \frac{1}{\epsilon}) \quad (14)$$

*where $N$ and $I$ are the batch size and the iteration number, $\zeta$ is the disagreement coefficient related to the learning problem defined in Appendix E.2.3, and $d$ is the VC-dimension (Vapnik, 1998) of the reward model defined in Appendix E.1.*

The above theorem shows that the proposed HSBC algorithm can achieve Probably Approximately Correct (PAC) learning of the human reward. The learned reward function $r_{\boldsymbol{\theta}}$ can make preference prediction with an arbitrarily low error rate $\epsilon$ under an arbitrary high probability $1 - \delta$, under the human preference data complexity in (14). In practical scenarios, the reward function $r_{\boldsymbol{\theta}}$ is often represented

by a deep neural network. While Theorem 4.2 is stated in terms of VC-dimension, the measure is applicable to many neural network classes. For instance, in multilayer perceptrons (MLPs) with ReLU activations and bounded weights, the VC-dimension is finite and scales with the number of parameters and layers (Bartlett et al., 2019).

# 5. Robust Alignment

In this section, we extend the HSBC algorithm to handle false human preference data. We will establish a provably robust learning process, such that the agent can still learn $\theta_H$ regardless of unknown false human preferences.

First, for a false human preference $(\boldsymbol{\xi}^0, \boldsymbol{\xi}^1, y^{\text{false}})$, it is easy to verify that (3) will lead to

$$\theta_H \notin \{\boldsymbol{\theta} \in \Theta | f(\boldsymbol{\theta}, \boldsymbol{\xi}^0, \boldsymbol{\xi}^1, y^{\text{false}}) \geq 0\} \quad (15)$$

This means that a false human preference will mistakenly exclude the true $\theta_H$. In the HSBC algorithm, suppose a batch $\mathcal{B}_m$ of human preference includes unknown false human preferences, i.e., $\exists j = 1, 2, ..., N, y_{m,j} = y_{m,j}^{\text{false}}$. We have the following lemma stating the failure of the method.

**Lemma 5.1.** *Following the HSBC Algorithm, if there exists one false preference in $m$-th batch $\mathcal{B}_m$ of human preferences, then $\theta_H \notin \mathcal{C}_m$ and for all $i > m$, $\theta_H \notin \Theta_i$.*

Geometrically, Lemma 5.1 shows that any false human preference in preference batch $\mathcal{B}_m$ will make the batch cut $\mathcal{C}_m$ mistakenly "cut out" $\theta_H$, i.e., $\theta_H \in \mathcal{C}_m$. In the following, we show that the HSBC Algorithm will become provably robust by a slight modification of (6) or (9).

## 5.1. Cutting with Conservativeness Level $\gamma$

Our idea is to adopt a worst-case, conservative cut to handle unknown false human preferences. Specifically, let *conservativeness level* $\gamma \in [0, 1]$ denote the ratio of the maximum number of false human preferences in a batch of $N$ human preferences. In other words, with $\gamma$, we suppose a preference batch has *at most* $\lceil \gamma N \rceil$ ($\lceil \cdot \rceil$ is the round-up operator) false preference. $\gamma$ is a worst-case or conservative estimate of the false preference ratio and is not necessarily equal to the real error rate, which is unknown. In practice, $\gamma$ can be treated as a hyperparameter to reflect varying levels of human irrationality, or it can be adapted online based on run-time estimation of human behavior. In this work, we use $\gamma$ as a constant hyperparameter for simplicity.

With the conservativeness level $\gamma$, one can change (9) into

$$\mathcal{C}_i = \{\boldsymbol{\theta} | V_i(\boldsymbol{\theta}) > \lfloor (1-\gamma)N \rfloor - 0.5\}, \quad (16)$$

where $\lfloor \cdot \rfloor$ is the round-down operator, and the indicator function of the batch cut $\mathcal{C}_i$ thus can be expressed as

$$\mathbb{1}_{\mathcal{C}_i}(\boldsymbol{\theta}) = \mathbf{H}(V_i(\boldsymbol{\theta}) - \lfloor (1-\gamma)N \rfloor + 0.5). \quad (17)$$

The following lemma states that with the above modification of $\mathcal{C}_i$, the HSBC algorithm can still achieve the provable robustness against false human preference.

**Lemma 5.2.** *With the conservativeness level $\gamma$ and replacing (9) with $\mathcal{C}_i$ in (16), the HSBC Algorithm will have $\theta_H \in \mathcal{C}_i$ and $\theta_H \in \Theta_i$, i=1,2,3...I, regardless of false human preference.*

The lemma means by simply replacing (9) with (16) while keeping other settings unchanged, the HSBC algorithm is robust against human false labels. (9) is a special case of (16) since when $\gamma = 0$, $\lfloor (1-\gamma)N \rfloor = \lfloor N \rfloor = N$. Geometric interpretation of Lemma 5.2 is given below.

## 5.2. Geometric Interpretation

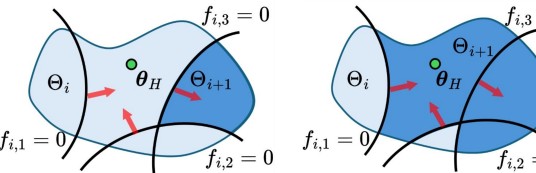

Figure 3: Geometry Interpretation of the robust HSBC with false preferences. Red arrows are the direction of the constraints, i.e., the $\boldsymbol{\theta}$ region satisfying $\{\boldsymbol{\theta} | f_{i,j}(\boldsymbol{\theta}) \geq 0\}$. Left: $y_{i,3}^{\text{false}}$ is false human preference, and thus simply taking the intersection will cut out $\theta_H$. Right: in robust HSBC, the regions with $V_i(\boldsymbol{\theta}) \geq 2$ are kept, thus $\theta_H$ is still contained in the new hypothesis space $\Theta_{i+1}$.

Fig. 3 shows a 2D geometric interpretation of the robust hypothesis cutting from $\Theta_i$ to $\Theta_{i+1}$ in the presence of false human preferences in batch $\mathcal{B}_i$. Here, the batch size is $N = 3$, and only the third preference $(\boldsymbol{\xi}_{i,3}^0, \boldsymbol{\xi}_{i,3}^1, y_{i,3}^{\text{false}})$ is false. We set $\gamma = 1/3$. As shown in Fig. 3, since $(\boldsymbol{\xi}_{i,3}^0, \boldsymbol{\xi}_{i,3}^1, y_{i,3}^{\text{false}})$ is a false preference, it will induce a constraint $f_{i,3}(\boldsymbol{\theta}) := f(\boldsymbol{\theta}, \boldsymbol{\xi}_{i,3}^0, \boldsymbol{\xi}_{i,3}^1, y_{i,3}^{\text{false}}) \geq 0$ which $\theta_H$ does not satisfy. As a result, simply taking the intersection of all constraints $\{\boldsymbol{\theta} | f_{i,j}(\boldsymbol{\theta}) \geq 0, j = 1, 2, 3\}$ will rule out $\theta_H$ by mistake. However, in this case, one can still perform a correct hypothesis space cutting (containing $\theta_H$) using (17). Replacing $N = 3$ and $\gamma = 1/3$ in (17), the indicator function for the batched cut is $\mathbb{1}_{\mathcal{C}_i}(\boldsymbol{\theta}) = \mathbf{H}(V_i(\boldsymbol{\theta}) - 1.5)$. This means by keeping the region in $\Theta_i$ with voting value $V_i(\boldsymbol{\theta})$ to be 2 or 3, $\theta_H \in \Theta_{i+1}$. In other words, by setting a mild voting threshold $\lfloor (1-\gamma)N \rfloor - 0.5$, a broader hypothesis space containing $\theta_H$ can be preserved.

# 6. Detailed Implementation

By viewing the case of all true human preferences as a special case with conservativeness level $\gamma = 0$, we present the implementation of the HSBC algorithm in Algorithm 1. The details are presented below.

## 6.1. Hypothesis Space Sampling

The step of hypothesis space sampling is to sample an ensemble of $M$ parameters, $\mathcal{E}_i = \{\boldsymbol{\theta}_i^k\}_{k=1}^M$, from the current

**Algorithm 1** Implementation for HSBC Algorithm

---

**Input:** Batch size $N$, ensemble size $M$, conservativeness level $\gamma$, disagreement threshold $\eta$, segment count per trajectory $Z$.

**for** $i = 0$ **to** $I$ **do**

Sample an assemble $\mathcal{E}_i$ from current hypothesis space (if $i = 0$, randomly initialize $\mathcal{E}_0$) with the method in Section 6.1. Filter and Densify $\mathcal{E}_i$ with the method in Section 6.2; 
*//Hypothesis Sampling*

Based on $\mathcal{E}_i$, generate trajectory using sampling-based MPC (Section 6.3). Select trajectory pairs based on disagreement score, and acquire a preference batch $\mathcal{B}_i$ (Section 6.3); 
*//Disagreement-based human query*

Update the sampling objective function with (19) and (20) using the new batch $\mathcal{B}_i$; 
*//Hypothesis space Cutting*

**end for**

**return** $\mathcal{E}_I$

---

hypothesis space $\Theta_i$. We let the initial hypothesis space $\Theta_0 = \mathbb{R}^r$, that is, $\mathbb{1}_{\Theta_0} = 1$, which guarantees possible $\boldsymbol{\theta}_H$ is contained in $\Theta_0$. In our implementation, we use neural networks as rewards, thus $\mathcal{E}_0$ is randomly initialized with common distributions for neural network initialization.

For $i > 0$, sampling $\mathcal{E}_i$ can be reformulated as an optimization with the indicator function (11), i.e.,

$$\mathcal{E}_i \subseteq \arg\max_{\boldsymbol{\theta}} \mathbb{1}_{\Theta_i}(\boldsymbol{\theta}) = \arg\max_{\boldsymbol{\theta}} \prod_{k=0}^{i-1} \mathbb{1}_{\mathcal{C}_k}(\boldsymbol{\theta}). \quad (18)$$

To use a gradient-based optimizer, (18) needs to be differentiable. Recall in (17) the only non-differentiable operation is $\mathbf{H}$. Thus, we use the smooth sigmoid function $\mathbf{sig}_\rho(\cdot)$ with temperature $\rho$ to approximate $\mathbf{H}$. In fact, other smooth functions with a similar shape can be used. From (17), $\mathbb{1}_{\mathcal{C}_i}$ will be replaced by the smooth approximation:

$$\hat{\mathbb{1}}_{\mathcal{C}_i}(\boldsymbol{\theta}) \approx \mathbf{sig}_\beta \left( \sum_{j=1}^{N} \mathbf{sig}_\alpha(f_{i,j}(\boldsymbol{\theta})) - \nu(1-\gamma)N \right), \quad (19)$$

where $\alpha, \beta, \nu$ are the tunable hyperparameters. Therefore, sampling $\mathcal{E}_i$ can be performed through

$$\mathcal{E}_i \subseteq \arg\max_{\boldsymbol{\theta}} \prod_{k=0}^{i-1} \hat{\mathbb{1}}_{\mathcal{C}_k}(\boldsymbol{\theta}). \quad (20)$$

To get diverse $\boldsymbol{\theta}$ samples in $\mathcal{E}_i$, we optimize (20) in parallel with diverse initial guesses. For $i > 0$, the optimization is warm-started using the previous $\mathcal{E}_{i-1}$ as initial values.

### 6.2. Sample Filtering and Densification

As $\mathcal{E}_i$ are solutions to a smoothed optimization (20), there is no guarantee that the samples are all in $\Theta_i$. Thus, we use the original indicator function $\mathbb{1}_{\Theta_i}(\boldsymbol{\theta})$ to filter the collected samples: remove $\boldsymbol{\theta}$s that are not satisfying $\mathbb{1}_{\Theta_i}(\boldsymbol{\theta})$. Since $\mathcal{E}_i$ after filtering may not have $M$ samples, we propose a densification step to duplicate some $\boldsymbol{\theta}$s (with also adding some noise) in $\mathcal{E}_i$ to maintain the same sample size $M$.

### 6.3. Reward Ensemble Trajectory Generation

With the ensemble $\mathcal{E}_i$ sampled from $\Theta_i$, the reward used for planning and control is defined by taking the mean value of predictions from the reward ensemble, i.e.

$$r_{\mathcal{E}_i}(\boldsymbol{s}, \boldsymbol{a}) = \mathbb{E}_{\boldsymbol{\theta} \in \mathcal{E}_i} r_{\boldsymbol{\theta}}(\boldsymbol{s}, \boldsymbol{a}). \quad (21)$$

In our implementation, we use sampling-based model predictive control (MPC) in the MJX simulator (Todorov et al., 2012) for trajectory generation. Half of the action plans generated from the MPC are perturbed with a decaying noise for exploration in the initial stage of learning. It is possible to use reinforcement learning for trajectory generation.

### 6.4. Disagreement Scoring and Query

In our algorithm, generated trajectories are stored in a buffer. When a new trajectory is generated, we segment it into $Z$ segments, and mix them with old segments for randomly picking segment pairs $(\boldsymbol{\xi}^0, \boldsymbol{\xi}^1)$ for disagreement test.

For any segment pair $(\boldsymbol{\xi}^0, \boldsymbol{\xi}^1)$, we measure their disagreement on the current hypothesis space using the reward ensemble $\mathcal{E}_i$. We define the following disagreement score:

$$\mathtt{DIS}_{\mathcal{E}_i}(\boldsymbol{\xi}^0, \boldsymbol{\xi}^1) = 4n_{\mathcal{E}_i}^+ n_{\mathcal{E}_i}^- / N^2 \quad (22)$$

where $n_{\mathcal{E}_i}^+$ is the number of $\boldsymbol{\theta}$s in $\mathcal{E}_i$ such that $J_{\boldsymbol{\theta}}(\boldsymbol{\xi}^0) > J_{\boldsymbol{\theta}}(\boldsymbol{\xi}^1)$; $n_{\mathcal{E}_i}^- = N - n_{\mathcal{E}_i}^+$. If no disagreement exist in $\mathcal{E}_i$, i.e., $\boldsymbol{\theta} \in \mathcal{E}_i$ makes one-sided judgment on $(\boldsymbol{\xi}^0, \boldsymbol{\xi}^1)$, $\mathtt{DIS}_{\mathcal{E}_i}(\boldsymbol{\xi}^0, \boldsymbol{\xi}^1) = 0$. When the disagreement is evenly split, the score is 1. We use a disagreement threshold $\eta \in (0, 1)$ to select valid disagreement pairs for human queries. A pair $(\boldsymbol{\xi}^0, \boldsymbol{\xi}^1)$ is rejected if $\mathtt{DIS}_{\mathcal{E}_i}(\boldsymbol{\xi}^0, \boldsymbol{\xi}^1) \leq \eta$; otherwise, the human is queried for a preference label $y$ for $(\boldsymbol{\xi}^0, \boldsymbol{\xi}^1)$. Trajectory generation and disagreement-based query are performed iteratively until $\mathcal{B}_i$ contains $N$ pairs.

## 7. Experiments

We test our method on 6 different tasks, as shown in Fig. 4:

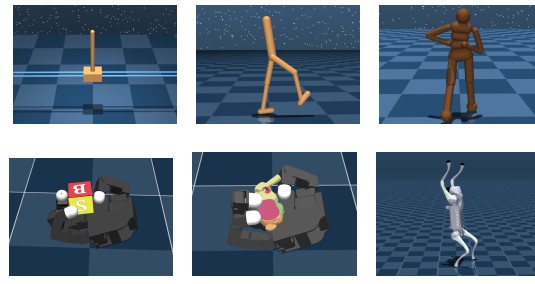

Figure 4: Task environments of experiments.

**3 dm-control tasks:** including Cartpole-Swingup, Walker-Walk, Humanoid-Standup. We directly use Cartpole, Walker and Humanoid to denote the tasks.

**2 in-hand dexterous manipulation tasks:** An Allegro hand in-hand reorients a cube and bunny object to any given

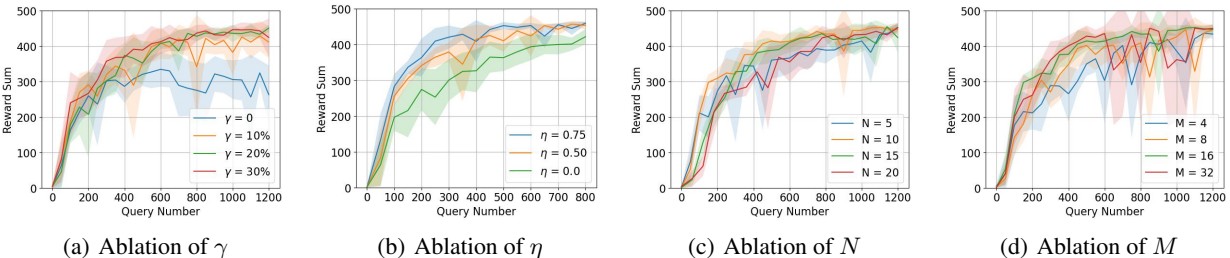

Figure 5: Learning curves for different tasks (rows) under different rates (columns) of false human preferences. The conservativeness level in HSBC Algorithm is the same as actual human false rate. All results are reported over 5 runs. The results show our method significantly outperforms the baseline when the false rate is high, and has a comparable performance when the false rate is 0 (no false preference).

Figure 6: Ablation study in Walker-Walk task for the choices of (a): conservativeness level $\gamma$, (b): disagreement threshold $\eta$, (c): batch size $N$, (d): ensemble size $N$. The learning curves are reported across 5 individual runs.

Table 1: Results on Tasks over 5 runs

| TASKS | ORACLE | FALSE RATE 0 % | | FALSE RATE 10% | | FALSE RATE 20% | | FALSE RATE 30% | |
|---|---|---|---|---|---|---|---|---|---|
| | | BASELINE | OURS | BASELINE | OURS | BASELINE | OURS | BASELINE | OURS |
| CARTPOLE | 148.6 | **143.3** $\pm$8.3 | 137.8 $\pm$7.9 | 101.4 $\pm$39.2 | **137.0** $\pm$8.5 | 52.3 $\pm$26.8 | **130.7** $\pm$2.0 | 42.8 $\pm$23.3 | **111.3** $\pm$16.8 |
| WALKER | 472.9 | 444.4 $\pm$13.4 | **459.4** $\pm$3.9 | 421.2 $\pm$20.6 | **441.4** $\pm$15.0 | 401.8 $\pm$37.6 | **447.1** $\pm$14.4 | 277.0 $\pm$62.3 | **417.2** $\pm$12.3 |
| HUMANOID | 210.1 | 107.0 $\pm$28.6 | **134.7** $\pm$30.5 | 85.6 $\pm$25.9 | **148.0** $\pm$14.9 | 84.2 $\pm$15.2 | **117.9** $\pm$19.9 | 69.3 $\pm$12.0 | **106.5** $\pm$27.0 |
| DEXMAN-CUBE | -0.139 | -0.281 $\pm$0.080 | **-0.207** $\pm$0.043 | -0.291 $\pm$0.092 | **-0.222** $\pm$0.057 | -0.360 $\pm$0.131 | **-0.238** $\pm$0.065 | -0.342 $\pm$0.024 | **-0.227** $\pm$0.058 |
| DEXMAN-BUNNY | -0.155 | -0.352 $\pm$0.096 | **-0.220** $\pm$0.038 | -0.356 $\pm$0.204 | **-0.278** $\pm$0.218 | -0.404 $\pm$0.101 | **-0.236** $\pm$0.026 | -0.505 $\pm$0.026 | **-0.263** $\pm$0.050 |
| GO2-STANDUP | -30.6 | -62.1 $\pm$22.9 | **-43.9** $\pm$4.2 | -84.1 $\pm$32.0 | **-36.7** $\pm$4.3 | -101.2 $\pm$18.0 | **-75.6** $\pm$21.7 | -85.5 $\pm$21.1 | **-80.5** $\pm$26.4 |

target. The two objects demand different reward designs due to the geometry-dependent policy. We use Dexman-Cube and Dexman-Bunny to denote two tasks.

**Quadruped robot stand-up task:** A Go2 quadruped robot learns a reward to stand up with two back feet and raise the two front feet. We use Go2-Standup to denote the task.

Detailed settings are in Appendix B.3. Similar to MJPC (Howell et al., 2022) but to better support GPU-accelerated sampling-based MPC, we re-implemented all environments in MJX (Todorov et al., 2012) with MPPI (Williams et al., 2015) as the MPC policy. Note that our method implementation is also applicable to other simulated environments, either by accepting slower policy inference in the absence of GPU parallelization, or by adapting alternative policy classes (e.g., RL-based policies) to generate trajectories.

**Simulated (false) human preference:** Similar to (Lee et al., 2021b; Cheng et al., 2024), a "simulated human" is implemented as a computer program to provide preference for the reward learning. The simulated human uses the ground-truth reward $r_{\theta_H}$ (see the ground truth reward of each task in Appendix B) to rank trajectory pairs. To simulate different rates of false human preferences, in each batch, a random selection of human preference labels is flipped.

**Baseline:** Throughout the experiment, we compare our method against a baseline implemented using the reward learning approach from PEBBLE (details in Appendix A).

**HSBC Algorithm settings:** In all tasks, rewards are MLP models. Unless specially mentioned, batch size $N = 10$, ensemble size $M = 16$, and disagreement threshold $\eta = 0.75$. In dm-control tasks and Go2-Standup, trajectory pairs of the first few batches are generated by a random policy for better exploration. Refer to Appendix C for other settings.

### 7.1. Results

For each task, we test our method and the baseline each for 5 independent runs with different random seeds. During the learning process, the reward ensemble $\mathcal{E}_i$ at the checkpoint iteration $i$ is saved for evaluation. The evaluation involves using the saved reward ensemble $\mathcal{E}_i$ to generate a new trajectory, for which the performance is calculated using the ground-truth reward. Each evaluation is performed in 5 runs with different random seeds. Evaluation performance at different learning checkpoints will yield the learning curve. We show the learning curves in Fig. 5. The mean and standard deviation across different runs are reported. To better

show the human query complexity, we set the x-axis as the number of queries, and the y-axis is the performance of $\mathcal{E}_i$ evaluated by ground-truth reward. Quantitatively, Table 1 shows the evaluation reward results of all tasks.

From both Fig. 5 and Table 1, we observe a comparable performance of our proposed method and the baseline for zero false human preferences. Both methods successfully learn reward functions to complete different tasks. As the rate of false preference increases, from 10% to 30%, we observe a significant performance drop of the baseline method. In contrast, the drop of the proposed method over a high rate of false human preference is not significant. These results demonstrate the robustness of the proposed method for reward learning against a high rate of false human preferences.

### 7.2. Ablation Study

**Conservativeness level** $\gamma$ Given a fixed rate of false human preference, We evaluate how different settings of conservativeness levels $\gamma = 0, 10\%, 20\%, 30\%$ affect the performance of the proposed method. Here, the actual human preferences have a fixed false rate of 20%. The ablation is tested for the Walker task, and the results over 5 runs are reported in Fig. 6(a). The results show that when $\gamma = 0$ (i.e., the algorithm aggressively cuts the hypothesis space without conservatism), the learning performance drops significantly. When $\gamma = 30\%$ exceeds the actual false rate (20%) but remains within a reasonable range, it has little impact on learning performance. However, we postulate that, in general, a higher $\gamma$ increases query complexity, as less of the hypothesis space is cut per iteration.

**Disagreement threshold** $\eta$ We evaluate the performance of our algorithm under different disagreement thresholds $\eta$ given all true human preferences. The test is in the Walker task. We set $\eta$ to 0, 0.5, 0.75 and show corresponding learning curves in Fig. 6(b). The results show that disagreement-based query can accelerate the learning convergence.

**Batch size** $N$ In Fig. 6(c), we show the reward learning performance for the Walker task with different choices of preference batch sizes, $N = 5, 10, 15, 20$, under a fixed rate 20% of false human preference. The results indicate a similar performance, suggesting that the learning performance is not sensitive to the setting of preference batch sizes.

**Ensemble size** $M$ In Fig. 6(d), we present the reward learning performance on the Walker task with varying ensemble sizes, $M = 4, 8, 16, 32$, under a fixed 20% rate of false human preferences and $\eta = 0.75$. The results show

Table 2: Comparison with multiple other methods under 20% and 30% false preference

| TASK | ORACLE | OURS | PEBBLE | RIME | SURF | MAE | t-CE |
|---|---|---|---|---|---|---|---|
| CARTPOLE-20% | 148.6 | **130.7** $\pm 2.0$ | 52.3 $\pm 26.8$ | 75.0 $\pm 45.3$ | 98.0 $\pm 35.8$ | 98.6 $\pm 25.9$ | 73.3 $\pm 16.3$ |
| CARTPOLE-30% | 148.6 | **111.3** $\pm 16.8$ | 42.8 $\pm 23.3$ | 81.0 $\pm 37.0$ | 62.3 $\pm 42.0$ | 59.9 $\pm 30.7$ | 52.0 $\pm 30.5$ |
| WALKER-20% | 472.9 | **447.0** $\pm 14.4$ | 401.9 $\pm 37.6$ | 408.4 $\pm 24.8$ | 397.2 $\pm 30.7$ | 425.5 $\pm 30.2$ | 410.8 $\pm 19.9$ |
| WALKER-30% | 472.9 | **417.2** $\pm 12.2$ | 277.0 $\pm 62.3$ | 310.2 $\pm 84.0$ | 292.0 $\pm 69.0$ | 288.3 $\pm 139.0$ | 345.6 $\pm 52.2$ |

that increasing the ensemble size can slightly improve the performance, but not significantly.

### 7.3. Comparison with Other Methods

We compare our HSBC algorithm against two state-of-the-art preference-based reward learning methods, SURF (Park et al., 2022) and RIME (Cheng et al., 2024), the PEBBLE baseline, and two other robust learning methods, MAE (Ghosh et al., 2017) and t-CE (Feng et al., 2021). All methods are evaluated on Cartpole and Walker tasks, with false preference rate of 20% and 30%, respectively. The comparison results are given in Table 2. Our HSBC method outperforms all of other methods in robust reward learning under high error rates. Among all those comparing methods, RIME excels at handling false preference labels with its label denoising design. In the Walker task, using t-CE loss also achieves robust results. The detailed experiment settings and the corresponding learning curves for each method are provided in the Appendix F.

### 7.4. Statistical Analysis of Learned Rewards

To evaluate the consistency between the learned and ground-truth rewards, we compute the Pearson correlation coefficient in three dm-control tasks under different false preference rates. For each task, we generate five 200-step trajectories and report the mean and standard deviation of the correlation values. The results, shown in Table 3, indicate strong correlations, suggesting that the learned rewards align well with ground-truth for high rates false human preference.

Table 3: Correlation between learned and ground-truth rewards.

| TASK | FALSE RATE - 0% | 10% | 20% | 30% |
|---|---|---|---|---|
| CARTPOLE | 0.928 $\pm 0.025$ | 0.888 $\pm 0.031$ | 0.914 $\pm 0.022$ | 0.851 $\pm 0.062$ |
| WALKER | 0.584 $\pm 0.035$ | 0.636 $\pm 0.060$ | 0.598 $\pm 0.070$ | 0.430 $\pm 0.062$ |
| HUMANOID | 0.673 $\pm 0.070$ | 0.657 $\pm 0.066$ | 0.546 $\pm 0.134$ | 0.500 $\pm 0.079$ |

### 7.5. Different False Preference Types

We further test our method under different types of irrational human preference labels (teachers) defined in B-Pref (Lee et al., 2021a), including "Stoc," "Mistake," and "Myopic" teachers (note we exclude "Equal" teacher, as ties are not considered in our formulation). The valuations are conducted on the Cartpole task with B-Pref hyperparameters set as $\beta = 10.0$ for Stoc, $\epsilon = 0.2$ for Mistake, and $\gamma = 0.98$ for Myopic. The conservativeness level of HSBC is fixed at 20%, and all other settings follow the original B-Pref setup. The results in Table 4 show our method handles "Mistake" and "Myopic" teachers well, but struggles with "Stoc" teachers. The reason could be that in the late stage of learning, the trajectories in a pair are close enough in reward values,

and thus "Stoc" teacher tends to provide very noisy labels, which hinders the convergence of the algorithm.

Table 4: Evaluation under different B-Pref teacher models.

| TEACHER | ORACLE | STOC | MISTAKE | MYOPIC |
|---|---|---|---|---|
| REWARD | 148.6 | 93.6 $\pm 30.7$ | 122.6 $\pm 14.6$ | 127.0 $\pm 26.5$ |

### 7.6. Evaluation on Real Human Data

We finally evaluate our HSBC algorithm using real human feedback on the CartPole and Walker tasks, with four human volunteers. We report the performance of the learned rewards across participants. To efficiently bootstrap learning, human feedback was collected after a small amount of simulated feedback—50 for CartPole and 100 for Walker. Each volunteer provided 50 preferences for CartPole and 100 for Walker. Due to the inherent noise of real human preferences, $\gamma$ was set to 40%. HSBC was compared against the PEBBLE baseline under identical conditions. The learning curves are presented in Fig. 7. The results demonstrate that HSBC achieves more stable convergence and superior performance when learning from real human feedback.

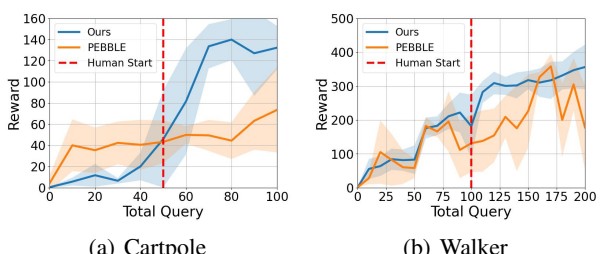

(a) Cartpole          (b) Walker

Figure 7: Reward learning with real human preference. HSBC outperforms baseline in reward performance and learning stability.

## 8. Conclusion

This paper proposes Hypothesis Space Batch Cutting (HSBC), a method for robust reward alignment from human preference that may include unknown false feedback. HSBC selects batches of human preference data based on disagreement and uses them to iteratively refine the hypothesis space of reward functions. Each preference batch leads to a voting function over the hypothesis space, which is then used to eliminate portions of the hypothesis space. To guard against false preference labels, a conservative cutting strategy is proposed to avoid overly aggressive hypothesis cutting. HSBC is geometrically interpretable and certifiable on human query complexity. Across diverse control tasks, HSBC matches the state-of-the-art methods with clean data and outperforms them under high false preference rates.

## Impact Statement

The HSBC method improves both interpretability and robustness in reward alignment by offering a geometric perspective on hypothesis space updates and a certifiable bound on human query complexity. Its conservative cutting strategy enables reliable learning even under a high rate of false preference labels, without requiring explicit identification of those labels. As a result, HSBC is resilient to noisy or adversarial human feedback and remains simple to implement, requiring no additional classification modules. These strengths make HSBC particularly well-suited for applications in robotics, autonomous systems, and human-in-the-loop decision-making, where false or inconsistent data are common and accurate alignment with human intent is critical. By enhancing robustness while preserving interpretability, HSBC provides a principled framework for preference-based learning in uncertain real-world environments.

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

## A. Baseline Method for Comparison

We present the details of the baseline method used for comparison. It is inspired by the method of PEBBLE (Lee et al., 2021b) on training the reward model, while uses the same setting to plan the trajectories and collect the preferences. The only differences between the baseline and our method are the ensemble size and the objective function. An ensemble of 3 reward models is maintained. After $i$ batches of preferences, the reward models are tuned to optimize the Bradley-Terry loss function of all previous preferences:

$$\mathcal{L} = \sum_{k=0}^{i-1} \sum_{j=0}^{N-1} (1 - y_{k,j}) P_{\boldsymbol{\theta}}^{BT}(\boldsymbol{\xi}_{k,j}^1 \succ \boldsymbol{\xi}_{k,j}^0) + y_{k,j} P_{\boldsymbol{\theta}}^{BT}(\boldsymbol{\xi}_{k,j}^1 \succ \boldsymbol{\xi}_{k,j}^0) \tag{23}$$

where

$$P_{\boldsymbol{\theta}}^{BT}(\boldsymbol{\xi}^0 \succ \boldsymbol{\xi}^1) = \frac{\exp \alpha_{base} J_{\boldsymbol{\theta}}(\boldsymbol{\xi}^0)}{\exp \alpha_{base} J_{\boldsymbol{\theta}}(\boldsymbol{\xi}^0) + \exp \alpha_{base} J_{\boldsymbol{\theta}}(\boldsymbol{\xi}^1)} \tag{24}$$

The parameter $\alpha_{base}$ is the temperature of the Bradley-Terry model. In the dexterous manipulation tasks, $\alpha_{base} = 0.5$. In all other tasks, $\alpha_{base} = 3$. In the experiments of cartpole swingup, $\alpha_{base} = 10$. The preferences are selected with disagreement threshold $\eta = 0.8$, which is similar to our approach.

## B. Detailed Description of the Tasks

### B.1. DM-Control Tasks

We re-implemented three dm-control tasks in MJX to perform sampling-based MPC, similar to (Howell et al., 2022).

**Cartpole**   We directly use the environment xml file in dm-control, with the same state and action definition in the original library. The goal is to swing up the pole to an upright position and balance the system in the middle of the trail. Use $\varphi, \dot{\varphi}$ to denote the angle and angular velocity of the pole, $x, \dot{x}$ to denote the horizontal position and velocity of the cart. The system is initialized with $x = 0$ and $\varphi = \pi$, along with a normal noise with standard deviation 0.01 imposed on all joint positions and velocities. $\boldsymbol{s} = (x, \sin\varphi, \cos\varphi, \dot{x}, \dot{\varphi})$ and $\boldsymbol{a}$ is a 1D control scalar signal for the horizontal force applied on the cart. We use a ground truth reward function:

$$r(\boldsymbol{s}, \boldsymbol{a}) = \text{upright}(\boldsymbol{s}) * \text{middle}(\boldsymbol{s}) * \text{small\_ctrl}(\boldsymbol{a}) * \text{small\_vel}(\boldsymbol{s}) \tag{25}$$

where $\text{upright}(\boldsymbol{s}) = (\cos\varphi + 1)/2$, $\text{middle}(\boldsymbol{s}) = \exp(-x^2)$, $\text{small\_ctrl}(\boldsymbol{a}) = (4 + \exp(-4\boldsymbol{a}^2))/5$, $\text{small\_vel}(\boldsymbol{s}) = (1 + \exp(-0.5\dot{x}^2))/2$.

**Walker**   We directly use the environment xml file in dm-control. The goal is to make the agent walk forward with a steady speed. The system is initialized at a standing position and a normal noise with standard deviation 0.01 is imposed on all joint positions and velocities. Use $z$ to denote the sum of $z-$coordinate of the torso and a bias $-1.2$, thus the value $z = 0$ when the walker stands up-straight. $\varphi_y$ to denote the torso orientation in $xz-$plane, $q$ to denote all joints between links and $\dot{x}$ to denote the torso linear velocity on $x-$axis. $\boldsymbol{s} = (z, \varphi_y, q, \dot{x}, \dot{z}, \dot{\varphi}_y, \dot{q})$ and $\boldsymbol{a} \in [0,1]^6$ stands for torques applied on all joints. We use a ground truth reward function:

$$r(\boldsymbol{s}, \boldsymbol{a}) = \frac{3 * \text{standing}(\boldsymbol{s}) + \text{upright}(\boldsymbol{s})}{4} * \text{move}(\boldsymbol{s}) \tag{26}$$

where $\text{standing}(\boldsymbol{s}) = \text{clip}(1 - |z|, 0, 1)$, $\text{upright}(\boldsymbol{s}) = (\cos\varphi_y + 1)/2$, $\text{move}(\boldsymbol{s}) = \text{clip}(\dot{x}, 0, 1)$.

**Humanoid**   We directly use the environment XML file in dm-control. The goal is to stand up from a lying position. The system is initialized at a lying position and a uniform noise between -0.01 and 0.01 is imposed on all joint positions and velocities. Use $z, quat$ to denote the $z-$coordinate and quaternion orientation of the humanoid torso. Use $v = [v_x, v_y, v_z], w = [w_x, w_y, w_z]$ to denote 3D linear and angular velocity of the torso. Use $q$ to denote all of the joints in the humanoid agent. $\boldsymbol{s} = [z, quat, q, v, w, \dot{q}]$ and $\boldsymbol{a} \in [0,1]^{17}$ stands for torques applied on all joints. We use a ground truth reward function:

$$r(\boldsymbol{s}, \boldsymbol{a}) = \text{standing}(\boldsymbol{a}) * \text{small\_vel}(\boldsymbol{s}) \tag{27}$$

where $\text{standing}(\boldsymbol{s}) = \text{clip}(z/1.2, 0, 1)$ and $\text{small\_vel}(\boldsymbol{s}) = \exp(-0.1 * \sqrt{v_x^2 + v_y^2} - 0.3 \|w\|)$.

## B.2. In-hand Dexterous Manipulation

We mainly focus on Allegro robot in-hand re-orientation of two different objects, a cube and a bunny. An allegro hand and an object are simulated in the environment. The allegro hand is tilted with quaternion $[0.0, 0.82, 0.0, 0.57]$. The objects have an initial position of $[0.0, 0.0, 0.05]$ with quaternion $[0, 0, 0, 1]$. At the beginning of every trajectory planning during training, a target orientation in the form of an axis-angle is randomly selected. The axis is uniformly randomly selected from $x, y, z$ axis and the angle is uniformly randomly selected from the set $\{\pm\frac{\pi}{4}, \pm\frac{\pi}{2}, \pm\frac{3\pi}{4}, \pi\}$. This angle-axis rotation is then converted to the target quaternion $q_{targ}$. The goal is to use the fingers of the robotic hand to rotate the object to the target orientation, as well as remain the object in the center of the hand. Use $q$ to denote the joint positions of the robot hand, $p$ to denote the object position, $\Delta quat$ to denote the quaternion different between current object pose and target pose ($\Delta quat = 1 - (q_{curr}^T q_{targ})^2$) and $ff_{tip}, mf_{tip}, rf_{tip}, th_{tip}$ to denote the positions of four fingertips (forefinger, middle finger, ring finger and thumb). The object and fingertip positions are multiplied by 10 to balance the scale. $\boldsymbol{s} = [q, p, \Delta quat, ff_{tip}, mf_{tip}, rf_{tip}, th_{tip}]$. $\boldsymbol{a} \in [0, 1]^{12}$ is the delta-position command on all joints and is multiplied by 0.2 to generate actual control signals. The ground truth reward is designed to be:

$$r(\boldsymbol{s}, \boldsymbol{a}) = -\text{cost\_quat}(\boldsymbol{s}) - 0.4\text{cost\_pos}(\boldsymbol{s}) - 0.05\text{cost\_contact}(\boldsymbol{s}) \tag{28}$$

Where $\text{cost\_quat}(\boldsymbol{s}) = \Delta quat$, $\text{cost\_pos}(\boldsymbol{s}) = \|p - (0.0, 0.0, 0.01) * 10\|^2$ and $\text{cost\_contact}(\boldsymbol{s}) = \|p - ff_{tip}\|^2 + \|p - mf_{tip}\|^2 + \|p - rf_{tip}\|^2 + \|p - th_{tip}\|^2$.

In evaluation, 6 trajectories are planned by setting different target of $\pm\frac{\pi}{2}$ rotation on $x, y, z$ axis. The mean reward per trajectory per time step is calculated and recorded.

## B.3. Go2-Standup

In this task, a go2 quadruped robot is simulated and the goal is to stand up on two back feet to keep the torso height on 0.6, and raise two front feet to the height of 1.2. The initial pose and the target pose are shown in Fig. 8. The system is initialized at a standing position on four legs, and a uniform noise between -0.01 and 0.01 is imposed on all joint positions and velocities. Use $z_t, quat$ to denote the $z-$coordinate and quaternion orientation of the robot torso. Use $z_{ff}, z_{rf}$ to denote $z-$coordinates of two front feet and two rear feet. To balance the scale, all features related to the $z$-coordinates are multiplied by 10. Use $v = [v_x, v_y, v_z], w = [w_x, w_y, w_z]$ to denote 3D linear and angular velocity of the torso. Use $q$ to denote all of the joint positions. $\boldsymbol{s} = [z_t, quat, q, z_{ff}, z_{rf}, v, w, \dot{q}]$ and $\boldsymbol{a} \in [0, 1]^{12}$ stands for the desired delta-position on all joints. We define the ground truth reward function as:

$$r(\boldsymbol{s}, \boldsymbol{a}) = -2.5\text{height\_cost}(\boldsymbol{s}) - 0.5\text{feet\_cost}(\boldsymbol{s}) - 10^{-6}\text{ang\_vel\_cost}(\boldsymbol{s}) - 10^{-2}\text{vel\_cost}(\boldsymbol{s}) \tag{29}$$

where $\text{height\_cost}(\boldsymbol{s}) = (z - 0.6)^2$, $\text{feet\_cost}(\boldsymbol{s}) = \|z_{ff} - 1.2\|^2 + \|z_{rf}\|^2$, $\text{ang\_vel\_cost}(\boldsymbol{s}) = \|w\|^2$ and $\text{vel\_cost}(\boldsymbol{s}) = \|v\|^2$.

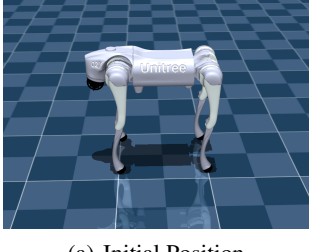

(a) Initial Position        (b) Stand-up Position

Figure 8: Illustration of the Go2-Standup task, the goal is to reach a stand-up posture with two feet.

## C. Model and Parameters in Experiment

**Reward Neural Networks** For all tasks, we use an MLP with 3 hidden layers of 256 hidden units to parameterize reward functions. The activation function of the network is ReLU. In the dm-control tasks, the input is a concatenated vector of state $\boldsymbol{s}$ and action $\boldsymbol{a}$. In dexterous manipulation and quadruped locomotion tasks, the input is purely the state $\boldsymbol{s}$ to perform

Table 5: Learning Parameter in Different Tasks

| TASK | $\alpha$ | $\beta$ | $Z$ | $I$ | $T$ | $T_{eval}$ | $T_{seg}$ |
|------|------|------|-----|--------|-----|--------|--------|
| CARTPOLE | 10.0 | 3.0 | 2 | 50/80 | 100 | 200 | 50 |
| WALKER | 5.0 | 3.0 | 2 | 80/120 | 150 | 500 | 50 |
| HUMANOID | 5.0 | 3.0 | 3 | 100/150 | 200 | 300 | 50 |
| DEXMAN-CUBE | 5.0 | 3.0 | 3 | 30/40 | 100 | 150 | 20 |
| DEXMAN-BUNNY | 5.0 | 3.0 | 3 | 30/40 | 100 | 150 | 20 |
| GO2-STANDUP | 5.0 | 3.0 | 3 | 80/120 | 150 | 100 | 25 |

Table 6: MPPI Parameter in Different Tasks

| TASK | Num | Hor | $\lambda$ | Std |
|------|-----|-----|-----------|-----|
| CARTPOLE | 256/512 | 20/25 | 0.01/0.01 | 1.0/0.75 |
| WALKER | 512/1024 | 25/30 | 0.01/0.01 | 1.0/0.75 |
| HUMANOID | 1024/1024 | 25/25 | 0.01/0.01 | 1.2/0.75 |
| DEXMAN-CUBE | 1024/1024 | 5/5 | 0.01/0.01 | 1.0/1.0 |
| DEXMAN-BUNNY | 1024/1024 | 5/5 | 0.01/0.01 | 1.0/1.0 |
| GO2-STANDUP | 1024/1024 | 25/25 | 0.005/0.01 | 1.0/0.75 |

faster learning. The output scalar value of the reward is scaled to the interval $[-1, 1]$ using the Tanh function. In particular, in the dexterous manipulation tasks, an intrinsic reward $-0.5\text{cost\_contact}(\boldsymbol{s})$ is imposed to the network output, forming a predicted reward to encourage grasping the object and bootstrapping the learning.

**Learning Settings**    The parameters $\alpha, \beta, Z, I$, trajectory length $T$, segment length $T_{seg}$ and evaluation trajectory length $T_{eval}$ slightly varies from the tasks, see Table 5. In the Table, the two number of batch number $I$ corresponds to the number with irrationality $\{0, 10\%\}$ and $\{20\%, 30\%\}$. The Adam (Kingma, 2014) optimizer parameter is identical for all dm-control and Go2-Standup tasks, the learning rate is set to be 0.005 with a weight decay coefficient 0.001. In dexterous manipulation tasks, the learning rate is 0.002 with a weight decay coefficient 0.001.

**MPPI Parameter**    The number of samples (Num), planning horizon (Hor), temperature ($\lambda$) and the standard deviation (Std) of the normal sampling distribution determine the quality of trajectories planned by MPPI. These parameters varies from different tasks, see Table 6. In the table, two numbers on each entry denotes the different settings in the training and evaluation stages.

## D. Proof of the Lemmas

### D.1. Proof of Lemma 4.1

*Proof.* By the definition of $y^{\text{true}}$ in (2),

$$f(\boldsymbol{\theta}, \boldsymbol{\xi}_{i,j}^0, \boldsymbol{\xi}_{i,j}^1, y_{i,j}^{\text{true}}) \geq 0 \tag{30}$$

combining the definition of $\mathcal{C}_i$ in (6), $\boldsymbol{\theta}_H \in \mathcal{C}_i, \forall i$. Hypothesis space $\Theta_i = \Theta_0 \cap \mathcal{C}_0 \cap \cdots \cap \mathcal{C}_{i-1}$. Combining $\boldsymbol{\theta}_H \in \Theta_0$ and $\boldsymbol{\theta}_H \in \mathcal{C}_i, \forall i, \boldsymbol{\theta}_H \in \Theta_i, \forall i$. $\square$

### D.2. Proof of Lemma 5.1

*Proof.* Without loss of generality, suppose $j$th preference in $m$th batch $(\xi_{m,j}^{(0)}, \xi_{m,j}^{(1)}, y_{m,j}^{\text{false}})$ has false label, Thus,

$$f_{m,j}(\boldsymbol{\theta}_H) < 0 \tag{31}$$

Which means $\boldsymbol{\theta}_H \notin \mathcal{C}_m$. With definition of $\mathcal{C}_m$ and $\Theta_i$, for all $i > m$,

$$\Theta_i \subseteq \Theta_m \subseteq \mathcal{C}_m \tag{32}$$

that leads to $\boldsymbol{\theta}_H \notin \Theta_i$ for all $i > m$. $\square$

## D.3. Proof of Lemma 5.2

*Proof.* With the definition of $\gamma$, there are at most $\lceil \gamma N \rceil$ incorrect labels and at least $\lfloor (1-\gamma)N \rfloor$ correct labels in one batch $\mathcal{B}_i$. For all $\boldsymbol{\theta}_H \in \Theta_H$, $f_{i,j}(\boldsymbol{\theta}_H) \geq 0$ holds for all tuples with correct labels $(\xi_{i,j}^0, \xi_{i,j}^1, y_{i,j}^{\text{true}})$. As a result, in the voting process of $V_i(\boldsymbol{\theta}_H)$ shown in (8), there will be at least $\lfloor (1-\gamma)N \rfloor$ "+1" votes and $V_i(\boldsymbol{\theta}_H) \geq \lfloor (1-\gamma)N \rfloor$. With the modified definition of $\mathcal{C}_i$ in (16), $\boldsymbol{\theta}_H \in \mathcal{C}_i$ and we have $\boldsymbol{\theta}_H \in \Theta_i$ □

# E. Proof of the Theorem 4.2

The proof of the theorem follows the pipeline in Chapter 6 in (Settles, 2012). In order for the paper to be self-contained, we include the complete proof here.

## E.1. Reward-Induced Classifier

The preference alignment can be viewed as a classification task. Use the new notation $x = (\boldsymbol{\xi}^0, \boldsymbol{\xi}^1)$ to denote a trajectory pair. This pair is fed to the classifier to predict a 0-1 label $y$ of which trajectory is better in human's sense. For any parameterized reward model $r_{\boldsymbol{\theta}}$, a classifier $h_{\boldsymbol{\theta}}$ can be induced by defining:

$$h_\theta(x) = \begin{cases} 0, J_{\boldsymbol{\theta}}(\boldsymbol{\xi}^0) \geq J_{\boldsymbol{\theta}}(\boldsymbol{\xi}^1) \\ 1, J_{\boldsymbol{\theta}}(\boldsymbol{\xi}^0) < J_{\boldsymbol{\theta}}(\boldsymbol{\xi}^1) \end{cases} \tag{33}$$

Recall that function $g$ is used to calculate the average reward on any trajectory. All of the classifiers induced by $r_\theta$ parameterized by $\theta \in \Theta_0$ forms a concept class $\mathcal{H} = \{h_\theta | \theta \in \mathbb{R}^r\}$. We assume the VC dimension of $\mathcal{H}$ is a finite number $d$.

Specifically, there is a human classifier correspond to $r_{\boldsymbol{\theta}_H}$ which defines the label $y$ corresponds to $x$:

$$y = h_H(x) = \begin{cases} 0, J_{\boldsymbol{\theta}_H}(\boldsymbol{\xi}^0) \geq J_{\boldsymbol{\theta}_H}(\boldsymbol{\xi}^1) \\ 1, J_{\boldsymbol{\theta}_H}(\boldsymbol{\xi}^0) < J_{\boldsymbol{\theta}_H}(\boldsymbol{\xi}^1) \end{cases} \tag{34}$$

Using $P_{XY}$ to denote the data-label joint distribution and use $P_X$ to define the marginal distribution of $x$ for simplicity. There exists one aligned classifier $h^* \in \mathcal{H}$ such that:

$$\forall x, y \sim P_{XY}, h^*(x) = y \tag{35}$$

Define the error rate of any $h \in \mathcal{H}$ as:

$$\text{err}(h) = P_X(h(x) \neq h^*(x)), \tag{36}$$

it is equivalent to express the preference prediction error rate $\text{err}(r_{\boldsymbol{\theta}}) = P(f(\boldsymbol{\theta}, \boldsymbol{\xi}^0, \boldsymbol{\xi}^1, y^{\text{true}}) < 0)$ in Theorem as $\text{err}(r_{\boldsymbol{\theta}}) = \text{err}(h_{\boldsymbol{\theta}})$.

## E.2. Definition of some Helper Functions, Sets and Coefficients

In this section we establish some definitions of helper functions, sets or coefficients, which are used in the main proof of Theorem 4.2.

### E.2.1. VERSION SPACE

Similar to the definition of $\mathcal{H}$, the classifier set generated by every $\Theta_i$ can be notes as $\mathcal{V}_i = \{h_{\boldsymbol{\theta}} | \boldsymbol{\theta} \in \Theta_i\}$, which is typically called *version space* in the context of active learning. By the definition of $\Theta_i$, for all $h \in \mathcal{V}_i$, $h$ can make perfect classfication of all data $(x, y)$ formed by the trajectory pairs and labels in all previous batches $\mathcal{B}_0, \ldots \mathcal{B}_{i-1}$. Also, since $\Theta_H \subseteq \Theta_i$, $h^* \in \mathcal{V}_i$ for all the time. Because the hypothesis space for parameters is shrinking, i.e., $\Theta_i \supseteq \Theta_{i+1}$, the version space is also shrinking such that $\mathcal{V}_i \supseteq \mathcal{V}_{i+1}$.

### E.2.2. DIFFERENCE BETWEEN CLASSIFIERS ON $\Xi$ AND CORRESPONDING BALLS

The data distribution provides a straightforward measure between two classifiers $h_1, h_2 \in \mathcal{H}$, by directly examining the proportion of data that they behave differently. A difference $\Delta(h_1, h_2)$ can be defined as the probability that two classifiers

behave differently, under the data distribution $\Xi$:

$$\Delta(h_1, h_2) = P_X(h_1(x) \neq h_2(x)) \tag{37}$$

With this different as a distance measure, an $\rho$−ball with radius $\rho$ and center $h_c$ can be defined as:

$$B(h_c, \rho) = \{h \in \mathcal{H} | \Delta(h_c, h) \leq \rho\} \tag{38}$$

### E.2.3. REGION OF DISAGREEMENT AND DISAGREEMENT COEFFICIENT

For a given version space $\mathcal{V}$ and use $\mathcal{X}$ to denote the instance space of $x$, the region of disagreement is a subset of the instance space:

$$\text{DIS}(\mathcal{V}) = \{x \in \mathcal{X} | \exists h_1, h_2 \in \mathcal{V} : h_1(x) \neq h_2(x)\} \tag{39}$$

With the definition of the region of disagreement, a *disagreement coefficient* was proposed by (Hanneke, 2007) to characterize the difficulty of active learning, which is defined with the expression

$$\zeta = \sup_{\rho > 0} \frac{P_X(\text{DIS}(B(h^*, \rho)))}{\rho} \tag{40}$$

The value $\zeta$ is determined by $\mathcal{H}$ and $\Xi$. The greater $\zeta$ is, the more the volume of the disagreement region of the $\rho$−ball around $h^*$ scales with $\rho$, which signifies greater learning difficulty.

### E.3. Passive PAC Sample Complexity Bound

Since the training set from the distribution $\Xi_{XY}$ is perfectly seperateble as in, using the famous probably approximately correct (PAC) bound for passive learning, with arbitary $\epsilon$ and $\delta$, $P(\text{err}(h) \leq \epsilon) \geq 1 - \delta$ can be achieved by perfectly fit training data with size $L_{PASS}$:

$$L_{PASS} \leq O(\frac{1}{\epsilon}(d\log(\frac{1}{\epsilon}) + \log\frac{1}{\delta})) \simeq O(\frac{d}{\epsilon}) \tag{41}$$

### E.4. Formal Proof of Theorem 4.2

*Proof.* For arbitary $\epsilon$ and $\delta$, define batch size $N$ to be:

$$N = \lceil c\zeta(d\log\zeta + \log\frac{1}{\delta'}) \rceil \tag{42}$$

with $c$ as a constant and a specially chosen $\delta'$ (the definition of which is shown later in the proof). With probability $1 - \delta'$ and for all $h \in \mathcal{V}_{i+1}$:

$$P_X(h(x) \neq h^*(x) | x \in \text{DIS}(\mathcal{V}_i)) \leq \frac{c'}{N}(d\log\frac{N}{d} + \log\frac{1}{\delta'}) \tag{43}$$

This is because data batch $\mathcal{B}_i$ is selected by disagreement (12). By the definition of $\mathcal{V}_{i+1}$, all of the data $\mathcal{B}_i$ comes from the set $\text{DIS}(\mathcal{V}_i)$ and all of the classifiers in $\mathcal{V}_{i+1}$ perfectly fits this batch. By switching the PAC bound $L_{PASS}$ in Appendix E.3 with $M$ and solving for $\epsilon$ we get the RHS of the inequality.

With proper choice of $c, c'$, $\log\frac{N}{d} \leq \log\zeta$ and the RHS of (43):

$$\frac{c'}{N}(d\log\frac{N}{d} + \log\frac{1}{\delta'}) \leq \frac{c'}{c\zeta} \leq \frac{1}{2\zeta} \tag{44}$$

As a result, for all $h \in \mathcal{V}_{i+1}$,

$$\text{err}(h) = P_X(h(x) \neq h^*(x)) \tag{45}$$
$$= P_X(h(x) \neq h^*(x) | x \in \text{DIS}(\mathcal{V}_i))P_X(x \in \text{DIS}(\mathcal{V}_j)) + \tag{46}$$
$$P_X(h(x) \neq h^*(x) | x \notin \text{DIS}(\mathcal{V}_i))P_X(x \notin \text{DIS}(\mathcal{V}_j)) \tag{47}$$

By the property $\mathcal{V}_{i+1} \subseteq \mathcal{V}_i$, $h \in \mathcal{V}_i$ and combining with $h^* \in \mathcal{V}_i$, $P_X(h(x) \neq h^*(x)|x \notin \text{DIS}(\mathcal{V}_i)) = 0$ and

$$\text{err}(h) \leq \frac{P_X(x \in \text{DIS}(\mathcal{V}_i))}{2\zeta} = \frac{P_X(\text{DIS}(\mathcal{V}_i))}{2\zeta} \tag{48}$$

This means $\mathcal{V}_{i+1} \subseteq B(h^*, \frac{P_X(\text{DIS}(\mathcal{V}_i))}{2\zeta})$, with the definition of the disagreement coefficient $\zeta$,

$$P_X(\text{DIS}(\mathcal{V}_{i+1})) \leq P_X(\text{DIS}(B(h^*, \frac{P_X(\text{DIS}(\mathcal{V}_i))}{2\zeta}))) \leq \frac{P_X(\text{DIS}(\mathcal{V}_i))}{2} \tag{49}$$

Thus, let $U_0 = P_X(\text{DIS}(\mathcal{V}_0))/2\zeta$, after $I = \log_2(U_0/\epsilon)$ batches of data and with $\delta' = \delta/I$, for all $h \in \mathcal{V}_I$, The relationship

$$\text{err}(h) \leq U_0 2^{-I} = \epsilon \tag{50}$$

holds with a union bound probability $1 - I\delta' = 1 - \delta$. With the definition of $\mathcal{V}_I$, for all $\theta \in \Theta_I$, $\text{err}(r_\theta) = \text{err}(h_\theta \in \mathcal{V}_I) \leq \epsilon$. Thus, the total number of the queries to perform PAC alignment can be expressed as:

$$K = IN = O(\zeta(d \log \zeta + \log \frac{\log 1/\epsilon}{\delta}) \log \frac{1}{\epsilon}) \tag{51}$$

This is the bound shown in Theorem 4.2 and completes the proof. $\qquad\square$

# F. Additional Experiments

## F.1. Comparison with Multiple Reward Learning Methods

The comparison includes two advanced preference-based reward learning methods, SURF (Park et al., 2022) and RIME (Cheng et al., 2024), as well as the PEBBLE (Lee et al., 2021b) baseline with two robust reward functions, MAE (Ghosh et al., 2017) and t-CE (Feng et al., 2021). For fairness in all comparisons (including the PEBBLE baseline in Section A), we replace the original RL policies in all methods with MPPI-based planners. The reward learning components remain consistent with their original implementations to ensure a fair evaluation of planning performance.

For the comparison of RIME (Cheng et al., 2024) and SURF (Park et al., 2022), we used the same settings as the original PEBBLE baseline for collecting trajectory segments. In RIME, KL-divergence between predicted preference probabilities and labels filters untrustworthy labels and flips them for improved learning. We set RIME parameters to $\alpha = 0.25$, $\beta_{\max} = 3.0$, $\beta_{\min} = 1.0$, $\tau_{\text{upper}} = -\ln(0.005)$, $k = 1/60$ for the Cartpole-Swingup task, and $\alpha = 0.3$, $\beta_{\max} = 2.2$, $\beta_{\min} = 1.7$, $\tau_{\text{upper}} = -\ln(0.005)$, $k = 1/100$ for the Walker-Walk task.

For SURF, we changed the length of collected segments to 60 and used temporal data augmentation to crop segments to a fixed length of 50. We choose $\tau = 0.95$, $\mu = 1.0$, and $\lambda = 1.0$ in both tasks. All algorithm parameters are selected to ensure the best performance of the baseline methods.

In MAE (Ghosh et al., 2017), the original loss function in PEBBLE baseline is replaced with $L_{\text{MAE}} = \mathbb{E}|\hat{y} - P_\theta|$ for robust reward learning. In t-CE (Feng et al., 2021), the loss is replaced with $L_{\text{t-CE}} = \mathbb{E} \sum_{i=1}^{t} \frac{1 - \hat{y}^\top P_\theta}{i}$. Here, $\hat{y}$ is the one-hot version of the noisy label, and $P_\theta$ is the predicted probability of human preference on trajectory pairs. We choose $t = 4$ in the t-CE loss based on its best performance.

The learning curves are shown in Fig. 9.

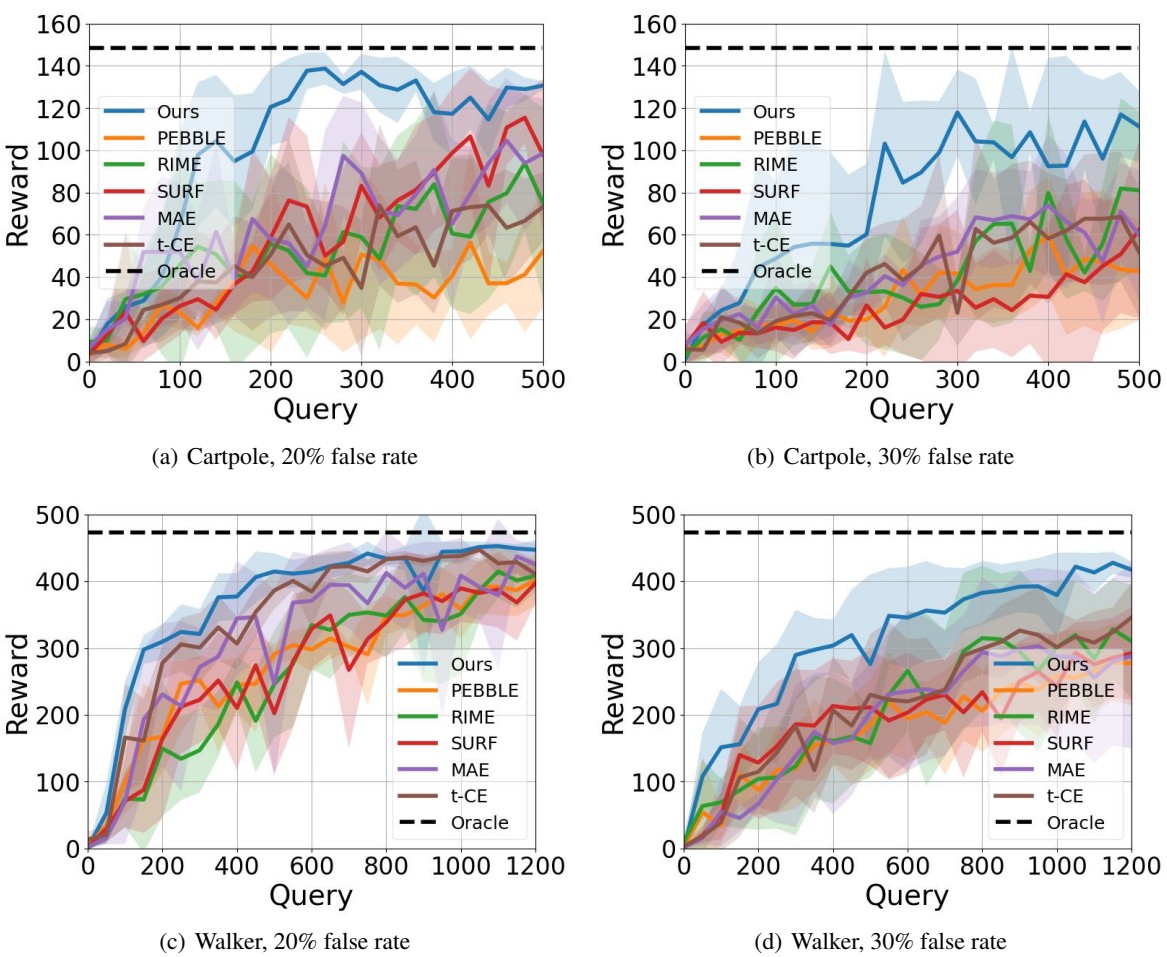

(a) Cartpole, 20% false rate

(b) Cartpole, 30% false rate

(c) Walker, 20% false rate

(d) Walker, 30% false rate

Figure 9: Learning curves of the methods used for comparison.

