# OpenReview forum: "Robust Reward Alignment via Hypothesis Space Batch Cutting"
_ICML.cc/2025/Conference — ICML 2025 poster_

### Official Review · Reviewer_GLNZ · 2025-03-14

**Overall Recommendation:** 4

**Summary:**

The paper introduces a novel method called Hypothesis Space Batch Cutting (HSBC), which iteratively refines a space of potential reward functions by using batches of human preferences to make "cuts" based on a voting function. To handle potentially erroneous human feedback, HSBC employs a conservative cutting method within each batch, ensuring robustness against false preferences while still learning the desired reward function.

**Claims And Evidence:**

Yes

**Essential References Not Discussed:**

N/A

**Experimental Designs Or Analyses:**

Well presented with good ablations, but lacks comparisons with baselines.

**Methods And Evaluation Criteria:**

It would be more informative to test the robustness of the method against different types of human errors as per Bpref [1]

1. B-pref: Benchmarking preference-based reinforcement learning." arXiv preprint arXiv:2111.03026 (2021).

**Other Comments Or Suggestions:**

1. The function f’s intuitive meaning and direct relationship to how a preference (true/false) constrains the hypothesis space could be emphasised better for clarity. Specifically, explaining why the $(1-2y)$ term effectively flips the sign of the difference in rewards based on the preference label would be helpful for a clearer understanding.

2. Below Eq 8, the Heaviside function seems to be described incorrectly

**Other Strengths And Weaknesses:**

The method seems interesting and novel and seems to improve performance. However, the set of baselines used is small. It could benefit from comparing with more baselines.

**Questions For Authors:**

1. The BPref paper cited in the paper shows there are multiple ways in which humans can be erroneous. How would HSBC deal with each of these individually? What kind of errors  can  it handle more easily?

2. With regards to the geometric interpretation, how does voting affect the cuts?

3. In practice, how much does the learned reward function (with HSBC) deviate from the ground truth?

4. Wouldn’t SURF (Semi-supervised Reward Learning with Data Augmentation for Feedback-efficient Preference-based Reinforcement Learning) also be a suitable baseline for comparison?

5. Is it necessary to use a sigmoid function in Eq 20 or would any smooth differentiable function be sufficient?

6. While the authors do a good job with the ablations, I wonder how the size of the ensemble M would affect the results.
Line 329 (left) ‘to using’

**Relation To Broader Scientific Literature:**

It could present an approach to improve the stability of preference based RL methods.

**Theoretical Claims:**

I hace not checked the proofs

---

> ### Author Rebuttal · Authors · 2025-04-01
>
> # Response to reviewer #GLNZ
>
> We sincerely thank you for your comments. Below, we address each of your comments in detail. All our responses will be incorporated into the final paper.
> ### 1. There are multiple ways... What kind of errors can it handle more easily?
> Thanks for your question. For different human error types in Bpref, our method cannot handle “Equal” preferences as we do not consider ties. However, our method can naturally handle “Skip” cases, as these can be simply excluded when constructing preference batches. We added evaluations on “Stoc,” “Mistake,” and “Myopic” teachers in the Cartpole task, following BPref hyperparameters: $\beta = 10.0$ for "Stoc," $\epsilon = 0.2$ for "Mistake," and $\gamma = 0.98$ for "Myopic". The conservativeness level was set to 20%, with other settings following the paper. Results confirm our method’s robustness across these feedback types.
> |Teacher|Oracle|Ours|Stoc|Mistake|Myopic|
> |-|-|-|-|-|-|
> |Reward|$148.6$|$130.7 \pm 2.0$|$93.6 \pm 30.7$|$122.6 \pm 14.6$|$127.0 \pm 26.5$|
>
> The results show our method handles “Mistake” and “Myopic” teachers well but struggles with “Stoc” teachers. This could be because in the late stage of learning, the trajectories are close enough in its rewards and “Stoc” teacher tends to provide very noisy labels, which hinders the convergence of the algorithm.
> ### 2. How does voting affect the cuts?
> Thanks for the question. We will use Figure 3 in the paper for illustration. The voting function $V_i (\boldsymbol{\theta})$ controls the aggressiveness of hypothesis space cuts. Retaining only the maximum-vote region $V_i (\boldsymbol{\theta})=N$ would remove all hypotheses disagreeing with any preference, assuming perfect feedback accuracy. However, human errors could wrongly eliminate the ground-truth reward (left panel of Figure 3). To address this, we set a mild vote threshold $V_i (\boldsymbol{\theta}) \geq \lfloor(1-\gamma) N\rfloor  -0.5$, preserving a broader hypothesis space containing $\boldsymbol{\theta}_H$ (Lemma 5.2), as shown in the right panel of Figure 3.
> ### 3. How much does the learned reward deviate from ground truth?
> Thanks for the question. To assess consistency, we compute the Pearson correlation between the learned and ground-truth rewards in Cartpole, Walker, and Humanoid. For each task, we generate five 200-step trajectories and report the mean and standard deviation. Results are below:
> |Task|false 0%|false 10%|false 20%|false 30% |
> |-|-|-|-|-|
> |Cartpole| $0.928 \pm 0.025$  | $0.888 \pm 0.031$  | $0.914 \pm 0.022$  | $0.851 \pm 0.062$ |
> | Walker| $0.584 \pm 0.035$  | $0.636 \pm 0.060$  | $0.598 \pm 0.070$  | $0.430 \pm 0.062$ |
> |Humanoid| $0.673 \pm 0.070$  | $0.657 \pm 0.066$  | $0.546 \pm 0.134$  | $0.500 \pm 0.079$  |
>
> The above results show that the learned reward and groundtruth reward has good correlation. Also, this correlation is weaker when the error rate increases, which is consistent with the performance results.
>
> ### 4. Comparison with SURF?
> Thank you for the comments. In this rebuttal, following the reviewer’s and all other reviewers’ comments, we have added the comparison with some other robust reward learning methods, including SURF. Please refer to the response of question 1 to reviewer #e9yz.
>
> ### 5. Is it necessary to use a sigmoid function in Eq 20...?
> No, any smooth differentiable function with a similar shape to sigmoid can be used, for example, the tanh function.
> ### 6. Ablation of the ensemble size M
> We added an ablation experiment on ensemble size $M$ on the walker-walk task with 20% error rate. The final result is shown  below:
> |$M$|4|8|16|32|
> |-|-|-|-|-|
> Reward|$433.9\pm 12.17$|$446.7\pm 14.3$|$447.0\pm 14.4$|$450.2\pm 9.22$
>
> With the groundtruth result of $472.9$. It can be seen from the result that increasing the ensemble size $M$ can slightly improve the performance, but not signfiicantly.
>
> ### 7. Better for clarity of the function f
> Thanks for the comment. The function $f$ represents the signed reward gap between ${\xi}^0$ and ${\xi}^1$, with the sign determined by human label $y$. Because $y$ is the “index” of the preferred trajectory, we set the sign as $1-2y$ to ensure $+1$ when $y=0$ and $-1$ when $y = 1$.
>
> As shown in Equ. (6), one human preference $({\xi}^0_{i,j}, {\xi}^1_{i,j}, y_{i,j})$, no matter true or false, imposes a constraint on hypothesis space using the inequality $f({\theta}, {\xi}^0_{i,j}, {\xi}^1_{i,j}, y_{i,j}) \geq 0$.
>
> ### 8. Typo of Heaviside function
> Thank the reviewer for point this out. This was a typo. The correct notation is $\mathrm{\mathbf{H}}(x) =1$ if $x \ge 0$ and $\mathrm{\mathbf{H}}(x) = 0$ otherwise. We will fix this in the revised version.
>
> ### Reference
> [1]Lee, Kimin, et al. “B-pref: Benchmarking preference-based reinforcement learning.” arXiv preprint arXiv:2111.03026 (2021).

---

> > ### Comment · Reviewer_GLNZ · 2025-04-08
> >
> > I thank the authors for their detailed responses. I have raised my score accordingly.

---

> > > ### Author Response · Authors · 2025-04-09
> > >
> > > We sincerely thank you for your thoughtful feedback and for recognizing our revisions—we truly appreciate your time and the improved assessment.

---

### Official Review · Reviewer_sofd · 2025-03-14

**Overall Recommendation:** 3

**Summary:**

This study addresses the challenge of reward design in reinforcement learning and proposes a robust and efficient preference-based reward alignment method, particularly for noisy human feedback. The method introduces a novel framework called "hypothesis space batched cutting," which iteratively refines the reward hypothesis space using batches of human preferences. The method employs a conservative cutting mechanism to ensure robustness when facing incorrect preferences. In error-free settings, the framework performs comparably to PEBBLE, while significantly outperforming other methods under high error rates in preferences.

**Claims And Evidence:**

yes

**Essential References Not Discussed:**

see weakness

**Experimental Designs Or Analyses:**

yes

**Methods And Evaluation Criteria:**

yes

**Other Comments Or Suggestions:**

I think this is a good paper, but the experiments are somewhat weak, and it lacks discussion and comparison with relevant literature. If the author can provide reasonable rebuttals, I am willing to raise the score.

**Other Strengths And Weaknesses:**

Advantages:
1. The paper is very clear, and the logic is easy to understand.
2. I can fully understand the motivation of this paper; performing effective RLHF learning on noisy data is very important issue.
3. The theoretical analysis is thorough, and the method makes sense.

Disadvantages:

1. Does gamma really play an adaptive role?
2. Although the method is highly sophisticated, the experiments are the weak point of this paper. The paper only compares with the PEBBLE algorithm, which is a very basic baseline algorithm, while recently there have been more methods with better performance. Can HSBC work effectively across most algorithms? It would be better to compare it with more powerful baselines.
3. Why are the experiments conducted in a custom environment from dmc? Most similar papers for online rlhf/pbrl use the B-Pref benchmark environment. Will you release these new environments as benchmarks?
3.5. Similar to the above point, I observed that a possible reason might be that the environment requires large-scale parallelism to support MPPI. Would this affect the practicality in other environments or the transferability to real-world tasks?
4. Another issue is the lack of discussion of other reinforcement learning methods with noisy human feedback in related work and experiments. Although this type of work is novel and important, there are still some papers that have conducted preliminary studies, such as: [1] Cheng J, Xiong G, Dai X, et al. RIME: Robust Preference-based Reinforcement Learning with Noisy Human Preferences[J].
[2] Li Y, Das S, Taylor M E. CANDERE-COACH: Reinforcement Learning from Noisy Feedback[J]. arXiv preprint arXiv:2409.15521, 2024.
[3] Xue W, An B, Yan S, et al. Reinforcement learning from diverse human preferences[J]. arXiv preprint arXiv:2301.11774, 2023.
[4] Yuan Y, Hao J, Ma Y, et al. Uni-rlhf: Universal platform and benchmark suite for reinforcement learning with diverse human feedback[J]. arXiv preprint arXiv:2402.02423, 2024.
5. Have you tried using real human feedback? Real human feedback should inherently contain a certain noise. You may refer to [4].

**Questions For Authors:**

-----

After rebuttal, I raised my score from 2 to 3.

**Relation To Broader Scientific Literature:**

see weakness

**Theoretical Claims:**

yes

---

> ### Author Rebuttal · Authors · 2025-04-01
>
> # Response to Reviewer #sofd
>
> We sincerely appreciate your thoughtful feedback and constructive comments on our paper. Below, we address each of your concerns in detail. All our responses will be incorporated into the final paper.
>
> ### 1. Does gamma really play an adaptive role?
> Thank you for the comments. Currently, $\gamma$ is fixed as a (conservative) estimate of the batch error rate, but our method could be extended to adaptively adjust $\gamma$ if real-time error estimation is available, which we leave for future work.
>
> ### 2.  It would be better to compare it with more powerful baselines
> Thank you for the comments. In this rebuttal, following the reviewer’s and all other reviewers’ comments, we have added the comparison with some other robust reward learning methods. Please refer to the comparison results in our response of question 1 to reviewer #e9yz.
>
> ### 3. Why are the experiments conducted in a custom environment from dmc? ...
> Thank you for the comments. We chose to implement our approach in new benchmark environments because we need a GPU-accelerated simulation to accelerate MPPI policies with parallel sampling. All of our custom testing environments will be released to support future research and standardized benchmarking in this area.
>
> Regarding practicality in other simulated environments, the proposed method is directly applicable. Environments without GPU support or parallel simulation may limit the speed of MPPI-based policies. In addition, as we pointed in Section 6.3, it is possible to use other policies (like RL) to generate trajectories for comparison.
>
> As for real-world transferability, recent works such as [5] have demonstrated that sampling-based control policies with parallelized simulation backends like MJX can be effectively transferred to real-world robotic tasks.
>
> ### 4. Lack of discussion of other RL methods with noisy human feedback in related work and experiments.
>
> Thank you for the comments. Both RIME [1] and CANDERE-COACH [2] improve learning from noisy labels by filtering human feedback—RIME uses KL-divergence to filter and flip corrupted labels, while CANDERE-COACH trains a neural classifier to predict preferences and filters based on discrepancies with real labels. Xue’s work [3] employs an encoder-decoder structure for reward models, estimating confidence levels from latent distributions and performing a weighted average for better predictions. Unlike these methods, ours does not explicitly assess feedback quality but instead updates the hypothesis space conservatively based on entire preference batches. Uni-RLHF [4] introduces an annotation platform and large-scale feedback dataset, using accuracy thresholds and manual verification, which are impractical for online reward learning due to the absence of ground truth and the high cost of manual inspection.
>
> In the revised version, we will include the above discussion.
>
> ### 5. Have you tried using real human feedback?
> Thank you for the comments. We conducted a new experiment evaluating HSBC with real human feedback on CartPole and Walker. Four volunteers provided trajectory preferences, which were used for reward learning. HSBC ($\gamma=0.4$) was compared to PEBBLE under the same settings as in the main paper, with a small amount of simulated feedback for pretraining. The results (sum of reward), presented below, show performance across different numbers of human preferences, with the first table for CartPole and the second for Walker.
> |# human preferences|0|10|20|30|40|50|Oracle|
> |-|-|-|-|-|-|-|-|
> |Ours (reward)|$45.1 \pm 44.7$|$80.5 \pm 50.8$|$132.8 \pm 19.8$|$147.4 \pm 11.1$|$132.5 \pm 32.8$|$133.2 \pm 31.2$|$148.6$|
> |PEBBLE (reward)|$43.3 \pm 20.2$|$49.9 \pm 8.3$|$49.4 \pm 14.6$|$44.5 \pm 17.3$|$63.2 \pm 27.3$|$73.5 \pm 40.2$|$148.6$|
>
> |# human preferences|0|20|40|60|80|100|Oracle|
> |-|-|-|-|-|-|-|-|
> |Ours (reward)|$181.7 \pm 92.3$|$309.0 \pm 33.4$|$301.9 \pm 38.4$|$310.3 \pm 41.7$|$331.4 \pm 41.2$|$356.2.2 \pm 66.0$|$472.9$|
> |PEBBLE (reward)|$131.3 \pm 85.2$|$154.2 \pm 78.9$|$175.5 \pm 126.2$|$326.5 \pm 15.2$|$200.1 \pm 108.5$|$177.9 \pm 123.6$|$472.9$|
>
> It can be shown that our method can achieve more stable convergence and superior performance when handling human feedback.
>
> ### Reference
> [1]Cheng, et al. "Rime: Robust preference-based reinforcement learning with noisy preferences." arXiv preprint arXiv:2402.17257 (2024).
>
> [2]Li Y, et al. CANDERE-COACH: Reinforcement Learning from Noisy Feedback[J]. arXiv preprint arXiv:2409.15521, 2024.
>
> [3]Xue W, et al. Reinforcement learning from diverse human preferences[J]. arXiv preprint arXiv:2301.11774, 2023.
>
> [4]Yuan Y, et al. Uni-rlhf: Universal platform and benchmark suite for reinforcement learning with diverse human feedback[J]. arXiv preprint arXiv:2402.02423, 2024.
>
> [5]Li, et al. "Drop: Dexterous reorientation via online planning." arXiv preprint arXiv:2409.14562 (2024).

---

> > ### Comment · Reviewer_sofd · 2025-04-07
> >
> > I’m very sorry for the late reply. I mistakenly used 'official comment' which is not visible to the author, I have changed it to a rebuttal response. I realized this and hope everything is still in time. Thank you for the response. Most of my concerns have been addressed, but I still have a few minor questions: As a planning and sampling technique, MPPI introduces additional costs. Does the stability of HSBC come from this additional planning? Since methods similar to MPPI are not very effective in environments that do not support large-scale parallelization, even though I believe that methods based on Isaac Gym represent the direction for the future, none of the other baselines consider parallel sampling setting. Would this kind of comparison introduce new unfairness?

---

> > > ### Author Response · Authors · 2025-04-07
> > >
> > > # Further Response to Reviewer #sofd
> > >
> > > Dear Reviewer #sofd,
> > >
> > > We appreciate the reviewer’s acknowledgement to our response and are glad to see the major concern is addressed. We sincerely thank for your thoughtful and constructive reply. Here are some further clarifications about your concerns on the usage of MPPI in our paper.
> > >
> > > 1. For fairness in all comparisons, including those presented in our paper, we replace the original RL policies in all baseline methods with MPPI-based planners. The reward learning components remain consistent with their original implementations to ensure a fair evaluation of planning performance.
> > >
> > > 2. We adopt MPPI as our planner instead of training an RL policy from scratch, as it avoids the need for extensive policy learning. Since our primary focus is on reward learning, the choice of planner is orthogonal: any controller capable of generating trajectories according to reward functions (e.g., sampling-based MPC like MPPI or RL) can be used as we discussed in the paper. Notably, several recent works ([1], [2]) have also employed simulator-based MPC as the planner in reward learning frameworks.
> > >
> > > 3. As shown in prior works (e.g., [3], [4], [5]), many successful reward learning methods employ non-parallelized simulators as predictive models within MPC frameworks. While our approach could similarly be extended to such settings, the current implementation leverages advanced parallelized environments like MJX for improved efficiency. Our method is also compatible with other GPU-parallel environments such as Isaac Sim, MuJoCo-Warp, and MuJoCo Playground.
> > >
> > > If the reviewer finds our clarifications and new results satisfactory, we would be grateful if the score could be updated to reflect the improvements and contributions more accurately.
> > >
> > > ## Reference
> > > [1] Yu, Wenhao, et al. "Language to rewards for robotic skill synthesis." arXiv preprint arXiv:2306.08647 (2023).
> > >
> > > [2] Liang, Jacky, et al. "Learning to learn faster from human feedback with language model predictive control." arXiv preprint arXiv:2402.11450 (2024).
> > >
> > > [3] Zakka, Kevin, et al. "Robopianist: Dexterous piano playing with deep reinforcement learning." arXiv preprint arXiv:2304.04150 (2023).
> > >
> > > [4] Li, Albert H., et al. "Drop: Dexterous reorientation via online planning." arXiv preprint arXiv:2409.14562 (2024).
> > >
> > > [5] Hess, Adrian, et al. "Sampling-Based Model Predictive Control for Dexterous Manipulation on a Biomimetic Tendon-Driven Hand." arXiv preprint arXiv:2411.06183 (2024).

---

### Official Review · Reviewer_e9yZ · 2025-03-14

**Overall Recommendation:** 3

**Summary:**

The paper introduces Hypothesis Space Batch Cutting (HSBC), a framework for robust reward alignment in reinforcement learning (RL). HSBC addresses the challenge of learning reward functions from human preferences, particularly in the presence of false or noisy feedback. The core idea is to iteratively refine a hypothesis space of reward models by "cutting" regions inconsistent with human preferences. Batches of preferences are queried based on disagreement among current hypotheses, and a voting function aggregates these preferences to determine the cuts. To handle errors, a conservative cutting strategy ensures that up to $\gamma$N false preferences per batch are tolerated. Theoretical guarantees include PAC learning bounds and robustness proofs. Empirical results demonstrate that HSBC outperforms representative methods like PEBBLE under high false preference rates (up to 30%) across diverse tasks (DM-Control, dexterous manipulation, quadruped locomotion).

**Claims And Evidence:**

The paper’s claims are generally supported by clear evidence.

**Essential References Not Discussed:**

N/A

**Experimental Designs Or Analyses:**

The experimental design is generally sound, but minor issues exist:
- Baseline Comparison: PEBBLE is used as the primary baseline, but comparisons with other robust Preference-based Reinforcement Learning (PbRL) methods (e.g., RIME [Cheng et al., 2024] or Xue et al. [2023]) would strengthen the evaluation.

**Methods And Evaluation Criteria:**

The methods and evaluation criteria are well-aligned with the problem:
- Hypothesis Space Cutting: The geometric interpretation of hypothesis updates is intuitive and addresses the limitations of prior preference-based RL methods (e.g., PEBBLE’s vulnerability to noise).
- Disagreement-Based Queries: Actively selecting trajectory pairs where current hypotheses disagree ensures efficient learning, as shown in the ablation study for $\eta$ (Figure 7b).
- Benchmark Tasks: The use of DM-Control, dexterous manipulation, and locomotion tasks (e.g., Go2-Standup) covers a broad range of control challenges, validating HSBC’s generality.

**Other Comments Or Suggestions:**

N/A

**Other Strengths And Weaknesses:**

**Strengths**
- The combination of batch cutting, voting functions, and conservatism provides a unique approach to robust PbRL.
- HSBC’s robustness is critical for real-world applications (e.g., human-robot interaction) where errors are inevitable.

**Weaknesses**
- All experiments are simulation-based. Validation in a practical setting would enhance credibility.

**Questions For Authors:**

1. Scalability to Neural Networks: Given that the theoretical bounds assume a finite VC-dimension, how does HSBC generalize to high-dimensional neural reward functions used in experiments? Could the bounds be adapted for neural models, or is this a limitation of the current framework?

**Relation To Broader Scientific Literature:**

The paper situates HSBC within the PbRL literature, contrasting with prior work on hypothesis space reduction (e.g., Sadigh et al. [2017]) and robust learning (e.g., Heo et al. [2025]). Key contributions include:
- Conservative Voting: A novel approach to handle false preferences without prior distribution assumptions, unlike methods like mixup [Heo et al., 2025].

**Theoretical Claims:**

Theoretical claims are rigorously presented.

---

> ### Author Rebuttal · Authors · 2025-04-01
>
> # Response to Reviewer #e9yZ
>
> We sincerely appreciate your thoughtful feedback and comments on our paper. Below, we address each of your concerns in detail. All our responses will be incorporated into the final paper.
>
> ### 1. Comparisons with other robust PbRL methods?
> Thank you for highlighting this point. Following the reviewer’ and other reviewers’ comments, in this rebuttal we have added the comparison with some other robust reward learning methods, including RIME[1], SURF[2], MAE[3] and t-CE[4]. The comparison is performed on the cartpole-swingup and walker-walk tasks, with 20% and 30% error rate.
>
> For the comparison of RIME [1] and SURF [2], we used the same settings as the original PEBBLE baseline for collecting trajectory segments. In RIME, KL-divergence between predicted preference probabilities and labels filters untrustworthy labels and flips them for improved learning. We set RIME parameters to $\alpha = 0.25$, $\beta_{max} = 3.0$, $\beta_{min} = 1.0$, $\tau_{upper} = -\ln(0.005)$, $k = 1/60$ for cartpole-swingup task and $\alpha = 0.3$, $\beta_{max} = 2.2$, $\beta_{min} = 1.7$, $\tau_{upper} = -\ln(0.005)$, $k = 1/100$ for walker-walk task. For SURF, we changed the length of collected segments to 60 and use temporal data augmentation to crop segments of a fixed-length 50. We choose $\tau = 0.95$, $\mu = 1.0$ and $\lambda=1.0$ for SURF in both tasks. All algorithm parameters are chosen for the best performance of the baseline methods.
>
> In  MAE[3], the original loss function is replaced with $L_{MAE} = \mathbb{E}|\hat{y} - P_{\theta}|$ for robust reward learning. In t-CE[4], the loss is replaced with $L_{t-CE} = \mathbb{E}\sum_{i=1}^t \frac{(1-\hat{y}^T P_{\theta})}{i}$. Here, $\hat{y}$ is the one-hot version of noisy label and $P_{\theta}$ as the predicted probability of human preference on trajectory pairs. We choose $t=4$ in t-CE loss of its best performance.
>
> The result (sum of reward) is shown in the table below:
>
> | Task|Oracle|Ours|PEBBLE|RIME|SURF|MAE|t-CE|
> |-|-|-|-|-|-|-|-|
> | Cartpole-Swingup-20% |  148.6  | $130.7 \pm 2.0$  | $52.3 \pm 26.8$  | $75.0 \pm 45.3$  | $98.0 \pm 35.8$  | $98.6 \pm 25.9$  | $73.3 \pm 16.3$  |
> | Cartpole-Swingup-30% |  148.6    | $111.3 \pm 16.8$  | $42.8 \pm 23.3$  | $81.0 \pm 37.0$  | $62.3 \pm 42.0$  | $59.9 \pm 30.7$  | $52.0 \pm 30.5$  |
> | Walker-Walk-20%    |  472.9    | $447.0 \pm 14.4$  | $401.9 \pm 37.6$  | $408.4 \pm 24.8$  | $397.2 \pm 30.7$  | $425.5 \pm 30.2$  | $410.8 \pm 19.9$  |
> | Walker-Walk-30%    |  472.9      | $417.2 \pm 12.2$  | $277.0 \pm 62.3$  | $310.2 \pm 84.0$  | $292.0 \pm 69.0$  | $288.3 \pm 139.0$  | $345.6 \pm 52.2$  |
>
> The results show that the proposed HSBC method outperforms baselines in robust learning under high error rates. Among the baselines, RIME excels at handling false preference labels with its label denoising design. In the Walker task, using t-CE loss also achieves robust learning.
>
> We will include the above baseline comparison in the revised version of the paper.
>
>
> ### 2.  Validation in a practical setting?
> We appreciate this observation. To address this, we performed HSBC on real-human feedback, please refer to the response of question 5 to reviewer #sofd.
>
> ### 3.The theoretical bounds assume a finite VC-dimension, how does HSBC generalize to high-dimensional neural reward functions?
> We thank the reviewer for the helpful comments. The VC-dimension is finite for certain neural network classes, such as multilayer perceptrons (MLPs) with ReLU activations, where it scales with the number of parameters and layers [5]. For such networks, the upper bound of the sample complexity applies. However, the exact VC-dimension of general neural networks, especially deep architectures with complex connectivity and unbounded weight norms, remains an open problem. In these cases, Theorem 4.2 provides a worst-case upper bound, which may be conservative. Tighter complexity bounds are an important direction for future work.
>
> We appreciate your time and effort in reviewing our work. Your feedback has been invaluable in strengthening our manuscript.
>
> ### Reference
> [1]Cheng, et al. "Rime: Robust preference-based reinforcement learning with noisy preferences." arXiv preprint arXiv:2402.17257 (2024).
>
> [2]Park, et al. "Surf: Semi-supervised reward learning with data augmentation for feedback-efficient preference-based reinforcement learning." arXiv preprint arXiv:2203.10050 (2022).
>
> [3]Ghosh, , et al. "Robust loss functions under label noise for deep neural networks." Proceedings of the AAAI conference on artificial intelligence. Vol. 31. No. 1. 2017.
>
> [4]Feng, et al. "Can cross entropy loss be robust to label noise?." Proceedings of the twenty-ninth international conference on international joint conferences on artificial intelligence. 2021.
>
> [5]Bartlett, et al. "Nearly-tight VC-dimension and pseudodimension bounds for piecewise linear neural networks." Journal of Machine Learning Research 20.63 (2019): 1-17.

---

### Decision · Program_Chairs · 2025-05-01

**Decision:**

Accept (poster)

**Comment:**

The paper presents Hypothesis Space Batch Cutting (HSBC), a method for robust reward alignment in reinforcement learning (RL), aiming to address the learning reward functions from human preferences in the presence of false or noisy feedback. The core idea is to iteratively refine a hypothesis space of reward models by "cutting" regions inconsistent with batches of human preferences, using a voting function and a conservative cutting method to handle potentially erroneous feedback. The method is theoretically grounded with PAC learning bounds and robustness proofs.

Reviewers generally appreciated the clarity of the paper, the motivation, and the theoretical underpinnings. The primary initial concern was the limited scope of experimental comparisons, which the authors addressed in their rebuttal by including additional state-of-the-art robust PbRL methods. Reviewers were also interested in the method's performance with real human feedback and under different types of errors, which the authors also explored in their responses. While some questions regarding the experimental setup and the use of parallelized environments were raised, the authors provided clarifications.

One reviewer raised the score, while some reviewers were not responsive. The meta-review takes into the account the authors' responses to all the reviewers.